# Modelling the drivers of the spread of *Plasmodium falciparum hrp2* gene deletions in sub-Saharan Africa

Oliver J Watson[1]*, Hannah C Slater[1], Robert Verity[1], Jonathan B Parr[2], Melchior K Mwandagalirwa[2,3], Antoinette Tshefu[3], Steven R Meshnick[2], Azra C Ghani[1]

[1]Medical Research Council Centre for Outbreak Analysis and Modelling, Department of Infectious Disease Epidemiology, Imperial College London, London, United Kingdom; [2]Division of Infectious Diseases, University of North Carolina, Chapel Hill, United States; [3]Ecole de Santé Publique, Faculté de Medecine, Université de Kinshasa, Kinshasa, Democratic Republic of the Congo

**Abstract** Rapid diagnostic tests (RDTs) have transformed malaria diagnosis. The most prevalent *P. falciparum* RDTs detect histidine-rich protein 2 (PfHRP2). However, *pfhrp2* gene deletions yielding false-negative RDTs, first reported in South America in 2010, have been confirmed in Africa and Asia. We developed a mathematical model to explore the potential for RDT-led diagnosis to drive selection of *pfhrp2*-deleted parasites. Low malaria prevalence and high frequencies of people seeking treatment resulted in the greatest selection pressure. Calibrating our model against confirmed *pfhrp2*-deletions in the Democratic Republic of Congo, we estimate a starting frequency of 6% *pfhrp2*-deletion prior to RDT introduction. Furthermore, the patterns observed necessitate a degree of selection driven by the introduction of PfHRP2-based RDT-guided treatment. Combining this with parasite prevalence and treatment coverage estimates, we map the model-predicted spread of *pfhrp2*-deletion, and identify the geographic regions in which surveillance for *pfhrp2*-deletion should be prioritised.

DOI: https://doi.org/10.7554/eLife.25008.001

*For correspondence:
o.watson15@imperial.ac.uk

**Competing interests:** The authors declare that no competing interests exist.

## Introduction

Efforts to control malaria globally have made substantial progress in the last 15 years (*World Health Organization, 2015a*). This progress reflects the impact made by reinvigorated political commitment that has yielded a twenty-fold increase in international funding for the control and elimination of malaria (*World Health Organization, 2015a*). The World Health Organisation (WHO) Global Technical Strategy for Malaria 2016–2030 sets ambitious goals to further reduce incidence and mortality rates by 90% by 2030 (*World Health Organization, 2015b*). Central to achieving these goals is the need to test, treat and track all malaria (*World Health Organization, 2010*).

In sub-Saharan Africa (SSA), diagnostic testing of suspected malaria cases has risen from 36% to 60% between 2005 and 2014 (*World Health Organization, 2015a*). Microscopy was historically the most common method for diagnosis; however rapid diagnostic tests (RDTs) accounted for 71% of all diagnostic testing of suspected cases in 2014 (*World Health Organization, 2015a*). The most widely used RDTs target histidine-rich protein 2 (HRP2), which is expressed by the *Plasmodium falciparum* (Pf) specific gene *pfhrp2*, with over 85% of RDTs tested in the WHO Foundation for Innovative New Diagnostics (FIND) Malaria RDT Evaluation Programme targeting PfHRP2 (*World Health Organization, 2012a*).

**eLife digest** Since the turn of the millennium, a large increase in international funding has helped to reduce the public health impact of malaria. The introduction of rapid diagnostic tests has played a central role in these efforts, particularly in remote areas that are heavily affected by the disease. These tests analyse human blood samples for specific proteins that are produced by malaria parasites.

The most common rapid diagnostic tests for malaria detect a protein called HRP2, which is produced by the deadliest malaria parasite, *Plasmodium falciparum*. Recently, however, cases have emerged where the tests have failed to detect these malaria infections. The first occurred in South America, and were found to be because some malaria parasites no longer possessed the gene that produces HRP2. Since then, malaria parasites that lack this gene have been found in several locations in Africa. This raises the question of whether using the tests favours the survival and spread of parasites that cannot produce the HRP2 protein.

Using mathematical modelling techniques, Watson et al. now present evidence that suggests that the use of HRP2-detecting rapid diagnostic tests over the past 10 years could have favoured the evolution of malaria parasites that lack this protein. Furthermore, the models suggest that the conditions that are most likely to cause such selection are places where malaria infections are not common but people seek treatment at high rates.

Using this information, Watson et al. created a map of 160 locations in Africa most at risk of rapid diagnostic test-driven selection against the gene that produces HRP2. Public health authorities could use these maps to determine where they should more closely monitor malaria parasites to see if they lack this gene.

Future genetic investigations will be required in the high-risk areas to confirm and refine the predictions. The development of rapid diagnostic tests that detect other malaria proteins will also be essential if malaria parasites that lack HRP2 continue to spread.

DOI: https://doi.org/10.7554/eLife.25008.002

False-negative RDT results due to a partial or complete deletion of the *pfhrp2* gene have been reported in areas of South America since 2010, resulting in the recommendation against the use of PfHRP2-based RDTs in these areas (*Akinyi et al., 2013*; *Abdallah et al., 2015*; *Cheng et al., 2014*). These *pfhrp2*-deleted mutants may still possess a functioning *pfhrp3* gene; however, the cross reactivity between PfHRP2-based RDT antibodies and PfHRP3 epitopes is such that a positive result may only occur at very high parasitaemia (*Baker et al., 2005*). Confirmed *pfhrp2*-deleted mutants are rarer in Africa, with the first cases reported in Mali in 2012 (*Koita et al., 2012*). However, recently confirmed occurrences in Ghana, (*Amoah et al., 2016*) Zambia, (*Laban et al., 2015*) the Democratic Republic of Congo (DRC), (*Parr et al., 2016*) Rwanda (*Kozycki et al., 2017*) and Eritrea (*Berhane et al., 2017*) (*Table 1*) have prompted the WHO to host Technical Consultations on *pfhrp2/3*-deletions and to issue interim guidance for malaria control programs (*World Health Organization, 2017*; *World Health Organization, 2016a*; *World Health Organization, 2016b*). This raises the concern that *pfhrp2*-deleted mutants may be selected for by RDT-guided treatment decisions – which if confirmed would be one of the first example of selection of a pathogen through diagnostic testing.

Here we use mathematical modelling to characterise the impact of introducing PfHRP2-based RDTs on the emergence and spread of *pfhrp2*-deleted parasites. We adapt a previously published transmission model (*Griffin et al., 2016*), incorporating the transmission of *pfhrp2*-deleted mutants and the contribution of PfHRP3 cross-reactivity to identify settings in which the selective pressure favouring *pfhrp2*-deleted strains is greatest. In addition, we conduct sensitivity analyses to characterise the influence of assumptions within our model concerning adherence to RDT-guided treatment decisions, the use of microscopy-based diagnostic testing, fitness costs associated with the mutant parasite and the impact of non-malarial fevers upon the selective advantage of *pfhrp2* gene deletions. We continue by using a nationally representative cross-sectional study of *pfrhp2*-deletion in the DRC (*Parr et al., 2016*) to estimate the prevalence of *pfhrp2*-deleted mutants prior to RDT introduction. This, in turn, allows us to map predicted geographical regions across SSA where *pfhrp2-*

**Table 1.**
Published studies reporting *P. falciparum* in Africa with deletions or no deletions of the *pfhrp2* gene (*Cheng et al., 2014*).

| Origin | | Source of samples* | Initial evidence | | | Gene deletion analysis by PCR | | | Antigen analysis | | Ref | Prevalence (no. of samples, year of collection) |
|---|---|---|---|---|---|---|---|---|---|---|---|---|
| Country | Area | | Microscopy | Quality RDT | Species PCR | pfhrp2 (exon 1 and 2) | No. single copy genes | Flanking genes | HRP ELISA | 2nd quality RDT | | |
| Mali | Bamako | A/S | D | ND | D | D | 1 | ND | ND | ND | (*Koita et al., 2012*) | 2% (480, 1996) |
| DRC,Gambia, Kenya, Mozambique, Rwanda, Tanzania, Uganda | | S | D | ND | D | Exon 2 only | ND | ND | D | ND | (*Ramutton et al., 2012*) | 0% (77, 2–19 per country, 2005–2010) |
| Senegal | Dakar | S | D | ND | D | D | 1 | ND | ND | ND | (*Wurtz et al., 2013*) | 2.4% (136, 2009–2012) |
| Ghana | Accra and Cape Coast | A | D | D | D | Exon 2 only | 2 | ND | ND | ND | (*Amoah et al., 2016*) | 29.5% (315, 2015) |
| Zambia | Choma, South Zambia | A/S | D | D | D | D | 1 | ND | ND | ND | (*Laban et al., 2015*) | 20% (61, 2009–2012)[†] |
| DRC | Country-wide | A | D | D | D | D | 3 | D | ND | ND | (*Parr et al., 2016*) | 6.4% (783, 2013–2014) |
| Rwanda | Busogo, Musanze, Kayonza | S | D | D | D | Exon 2 only | 1 | ND | ND | D | (*Kozycki et al., 2017*) | 23% (140, 2014–2015) |
| Eritrea | Anseba, Debub, Gash-Barka, Northern Red-Sea | S | D | D | D | ND | 1 | ND | ND | D | (*Berhane et al., 2017*) | 80% (51, 2015) |

*Source of samples: S = Symptomatic case, A = Asymptomatic case, U = not specified, D = done; ND = not done.

[†] Authors suggested that failure to detect *pfhrp* gene in some samples was more likely to be the result of low parasite density rather than deletion

Note: Quality RDT indicates RDTs that meet the WHO RDT recommended procurement criteria based on WHO Malaria RDT Product Testing.

DOI: https://doi.org/10.7554/eLife.25008.003

deletion surveillance should be focused. These mapped predictions are explored across a range of estimates of the prevalence of *pfhrp2*-deleted mutants prior to RDT introduction.

## Results

Using our newly adapted model incorporating the transmission of *pfhrp2*-deleted mutants, we first explored the potential for RDT-guided treatment decisions to exert an evolutionary pressure on the prevalence of the mutant. *Figure 1* shows the predicted proportion of strains that are *pfhrp2*-deleted within the population after 10 years.

Within all settings that explored different transmission intensities and starting frequencies of *pfhrp2*-deletion, RDT introduction is predicted to increase the proportion of *pfhrp2*-deleted mutants. The strength of selection is predicted to be greatest at low PfPR (*Figure 1a*); however, a selective pressure is still predicted at both high PfPR and at low starting *pfhrp2*-deletion frequencies (*Figure 1b*). The variance in the selection pressure exerted by RDTs is also predicted to be greatest at low PfPR (*Figure 1c*). A more gradual but analogous trend is predicted in the proportion of the population that were only infected with *pfhrp2*-deleted mutants (*Figure 1d*). The prevalence of malaria within *Figure 1a* was also observed to increase after RDT introduction (*Figure 1—figure supplement 1*), with the greatest increase in lower transmission settings resulting from untreated infections due to false-negative RDT results.

Within the sensitivity analyses, a selective pressure is observed to exist at comparative fitness costs of greater than 90% (see *Figure 1—figure supplement 2*), however below this the *pfhrp2*-deletion allele is quickly lost. Both the introduction of additional diagnosis with microscopy-based methods and non-adherence to RDT results decreased the selective pressure, slowing the rate of *pfhrp2*-deletion emergence (see *Figure 1—figure supplement 3*). The introduction of non-malarial fevers, however, increased the rate of *pfhrp2*-deletion emergence (see *Figure 1—figure supplement 4*), even at 25% below the mean estimated rate of non-malarial fever. When these opposing factors were combined, RDT introduction is still predicted to increase the proportion of *pfhrp2*-deleted mutants (*Figure 1—figure supplement 5*).

The proportion of clinical cases seeking treatment (assumed here to be treated on the basis of an RDT result) is also predicted to exert a strong selection pressure for *pfhrp2*-deletion (*Figure 2*). A consistent relationship was seen across comparable PfPR ranges, with the lowest treatment seeking rates ($f_T$ = 0.2) yielding the slowest increase in the proportion of infections due to only *pfhrp2*-deleted mutants. Again, the lower PfPR categories show the greatest selection pressures for *pfhrp2*-deletion, with treatment seeking rates >30% and PfPR <25% leading to 20% of infections due to only *pfhrp2*-deleted mutants in fewer than five years (*Figure 2a*).

The selection pressure favouring *pfhrp2*-deletions is predicted to be weaker when PfHRP3 epitopes are assumed to cause positive RDT results (*Figure 2b*). In settings where PfHRP3 epitopes are assumed to cause a positive RDT result in 25% of cases ($\varepsilon$ = 0.25), there are four fewer prevalence categories that reach 20% of infections due to only *pfhrp2*-deleted mutants in fewer than five years. A similar effect is observed in the mean final frequency of *pfhrp2*-deletion, with 64% frequency recorded after 20 years when no PfHRP3 epitope effect is assumed in comparison to 56% when $\varepsilon$ is equal to 0.25 (*Figure 2—figure supplement 1*).

To estimate the starting frequency of *pfhrp2*-deleted mutants, we used estimates of the proportion of *pfhrp2*-deleted mutants from a national study in DRC (*Parr et al., 2016*) to calibrate the model. The calibration incorporated both the PfPR levels and estimates of the treatment rates in the 26 Divisions Provinciales de la Santé (DPS) that would drive selection of the mutant. We estimate a starting frequency of *pfhrp2*-deleted *P. falciparum* of 6% in the DRC prior to any introduction of RDTs. The observed relationship between the proportion of infections due to *pfhrp2*-deleted mutants and PCR PfPR among children 6–59 months of age (*Figure 3a*) displays a similar trend to the simulations, however with a notably steeper increase at lower prevalence. Of note, the same relationship was not predicted in the absence of selection pressure due to RDT-based treatments (i.e. purely on the basis of the variation in monoclonal infections) (*Figure 3b*).

Finally, using the baseline frequency estimate of 6% prior to RDT introduction, we explored 1000 different prevalence and treatment seeking rates spanning the range of estimates of the PfPR (*Bhatt et al., 2015*) and treatment levels across sub-Saharan Africa (SSA) in 2010 (*Cohen et al., 2012*) (*Figure 4—figure supplement 2*). The model output was aligned with these estimates by first

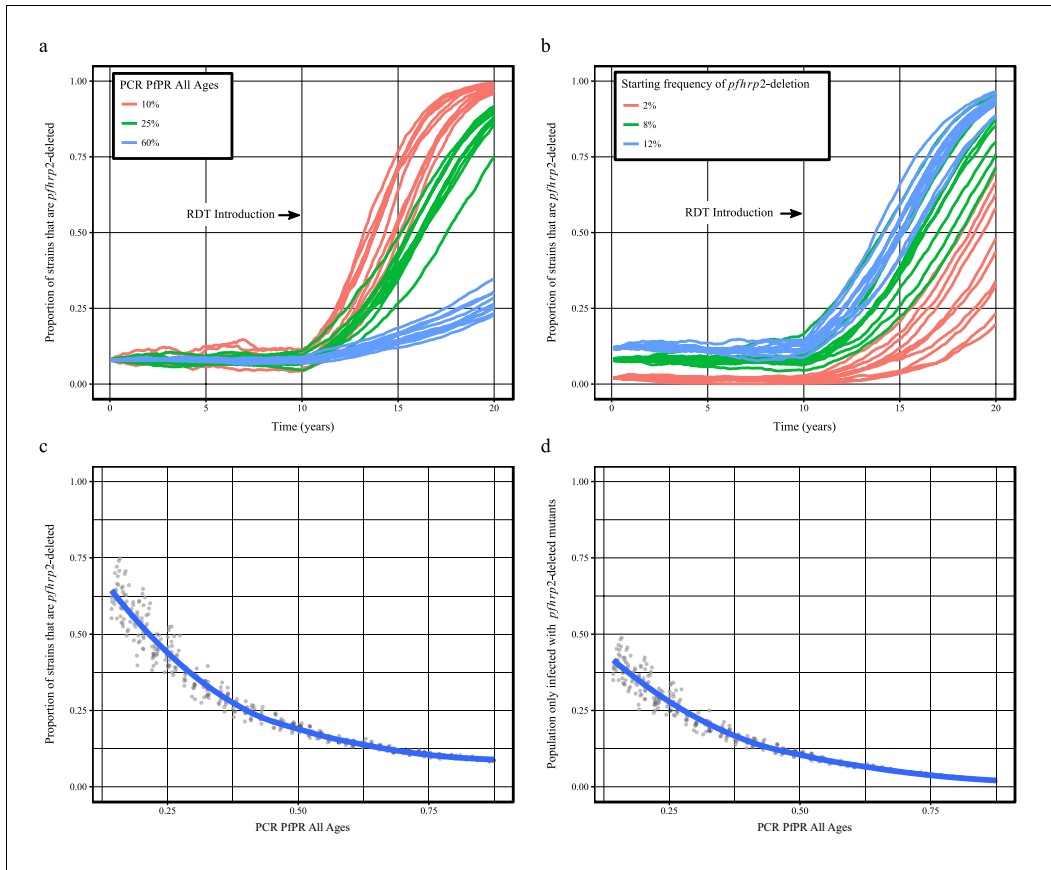

**Figure 1.** Predicted increase in *pfhrp2*-deletion upon RDT introduction after 10 years. Graphs show the time course of *pfhrp2*-deletion emergence under (**a**) different transmission intensities (10%, 25% and 60% PfPR) and 8% starting frequency of *pfhrp2*-deletion prior to RDT introduction and under (**b**) different assumed starting frequencies of *pfhrp2*-deletion prior to RDT introduction (2%, 8% and 12% starting frequency) and 25% PfPR. Five years after RDT introduction, the proportion of strains that are *pfhrp2*-deleted (**c**), and the proportion of the population that are infected with only *pfhrp2*-deleted mutants (**d**) is recorded. The dark grey dots denote individual simulation runs with a LOESS regression fit shown in blue. Source data for *Figure 1* is provided within *Figure 1—source data 1*.
DOI: https://doi.org/10.7554/eLife.25008.004

The following source data and figure supplements are available for figure 1:

**Source data 1.** Effect of transmission intensity and *pfhrp2*-deletion starting upon *pfhrp2*-deletion emergence.
DOI: https://doi.org/10.7554/eLife.25008.010

**Figure supplement 1.** Impact of increase *pfhrp2*-deletion upon malaria prevalence.
DOI: https://doi.org/10.7554/eLife.25008.005

**Figure supplement 2.** Impact of *pfhrp2*-deletion fitness cost.
DOI: https://doi.org/10.7554/eLife.25008.006

**Figure supplement 3.** Impact of microscopy use and non-adherence to RDT results.
DOI: https://doi.org/10.7554/eLife.25008.007

**Figure supplement 4.** Impact of non-malarial fever.
DOI: https://doi.org/10.7554/eLife.25008.008

**Figure supplement 5.** Combined impact of model assumptions.
DOI: https://doi.org/10.7554/eLife.25008.009

administrative units (*Figure 4—figure supplement 1*), which enabled us to project the potential increase of the mutant strain and its impact on RDT-guided treatment (*Video 1*). Our results suggest that 160 of 850 first-administrative regions may have over 20% of all infections due to only *pfhrp2*-deleted mutants by 2016 (*Figure 4c*). These areas, which we term of 'high HRP2 concern', are largely located in areas where $PfPR_{2-10}$ in 2010 was less than 25% (*Figure 4a*). A number of other regions,

classified as 'moderate HRP2 concern' have high treatment rates, and hence potential selective pressure, despite having comparatively higher transmission (*Figure 4b*). Our results also illustrate that regions with low transmission may have low HRP2 concern if the frequency of people seeking treatment is very low.

## Discussion

Our results demonstrate that the key drivers of *pfhrp2*-deletion selection are low malaria transmission and a high frequency of people seeking treatment and being correctly treated on the basis of diagnosis with a PfHRP2-based RDT. Based on Africa-wide estimates of parasite prevalence and treatment-seeking behaviour at the time of RDT-introduction, we identified 160 first-administrative units which we classify as 'high HRP2 concern'. These are areas where the *pfhrp2*-deleted strain is

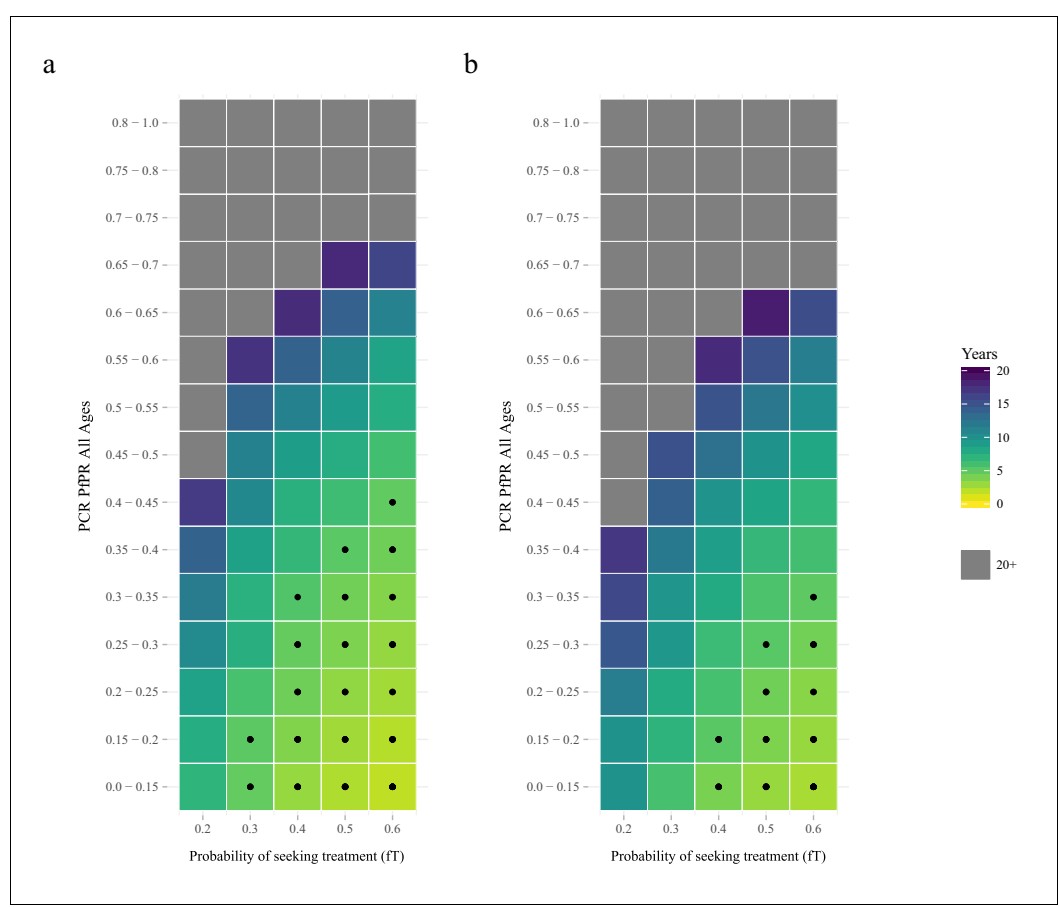

**Figure 2.** The predicted rate at which the population is only infected with *pfhrp2*-deleted mutants. The graphs show the time in years after RDT introduction at which 20% of the population are only infected with *pfhrp2*-deleted mutants up to a maximum follow-up time of 20 years post RDT introduction. PfHRP3 epitopes were assumed to cause a positive RDT result in (a) 0% or (b) 25% of individuals only infected with *pfhrp2*-deleted mutants. The plotted years represent the mean time grouped in each prevalence and treatment setting, with black dots representing where 20% was reached in less than five years. Each simulation had a starting *pfhrp2*-deletion frequency of 8% before RDT introduction. Source data for *Figure 2* is provided within *Figure 2—source data 1*.
DOI: https://doi.org/10.7554/eLife.25008.011

The following source data and figure supplement are available for figure 2:

**Source data 1.** Years after RDT introduction at which 20% of the population are only infected with *pfhrp2*-deleted parasites, with an assumed PfHRP3 epitope effect equal to 0% and 0.25%.
DOI: https://doi.org/10.7554/eLife.25008.013

**Figure supplement 1.** Frequency of *pfhrp2*-deletion after 20 years.
DOI: https://doi.org/10.7554/eLife.25008.012

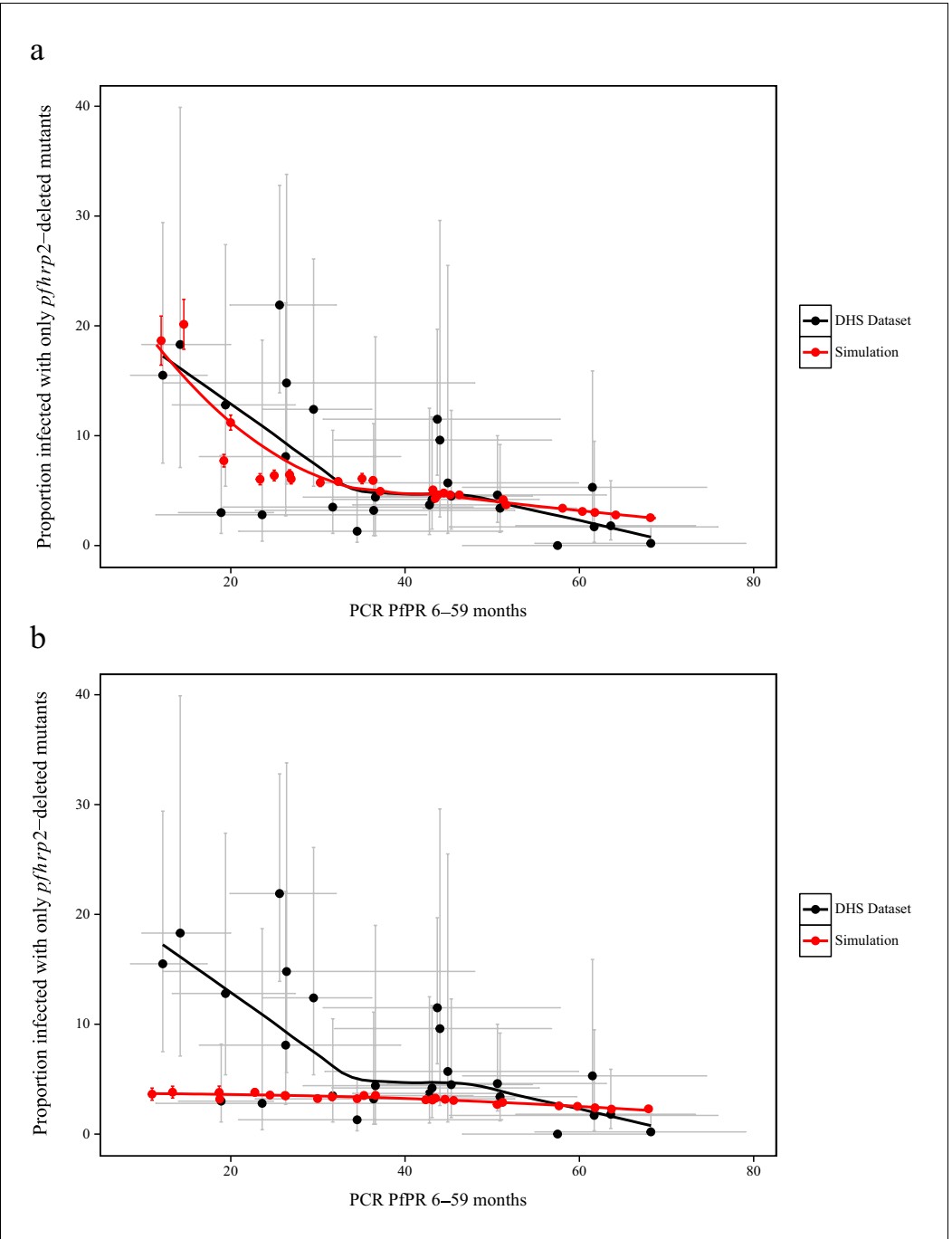

**Figure 3.** Simulated province level burden of *pfhrp2*-deleted mutants within the DRC, with an assumed probability of a clinical case seeking treatment, who is only infected with *pfhrp2*-deleted mutants, producing a positive RDT result ($\varepsilon$) equal to 0.25. In (**a**) the mean simulated proportion of children aged 6–59 months who are infected with only *pfhrp2*-deleted mutants is shown in red. Each region had an assumed starting frequency of 6% *pfhrp2*-deletion prior to RDT introduction in 2010 (2007 in North- and South-Kivu). The results in grey represent the recorded burden from the DHS survey (*Figure 3—source data 1*), with both datasets fitted with a LOESS regression. Error bars show the 95% confidence interval. In (**b**) the same simulation conditions were used as in (**a**) however it is assumed that no selection pressure is exerted by the introduction RDTs, i.e. $\varepsilon$ = 1. Source data for *Figure 3* is provided within *Figure 3—source data 1*.

DOI: https://doi.org/10.7554/eLife.25008.014

The following source data and figure supplement are available for figure 3:

**Source data 1.** Estimates of the proportion of pfhrp2-deleted mutants from a national study in DRC.

*Figure 3 continued on next page*

*Figure 3 continued*

DOI: https://doi.org/10.7554/eLife.25008.016

**Source data 2.** Simulated proportion of children aged 6–59 months who are only infected with *pfhrp2*-deleted parasites within the Democratic Republic of Congo, with an assumed PfHRP3 epitope effect equal to 0.25% and 1%, that is under no selection pressure.

DOI: https://doi.org/10.7554/eLife.25008.017

**Figure supplement 1.** Simulated province level burden of *pfhrp2*-deleted mutants within DRC, with an assumed probability of a clinical case seeking treatment, who is only infected with *pfhrp2*-deleted mutants, producing a positive RDT result (ε) equal to 0.

DOI: https://doi.org/10.7554/eLife.25008.015

expected to increase in frequency over a relatively short timescale, and hence where further surveillance efforts should be concentrated.

Our results are based on calibration to a large representative survey of malaria across DRC. Due to its size and location in the centre of SSA, the DRC is arguably one of the most representative countries for endemic malaria in Africa. That the model was able to predict the observed relationship in the DRC, despite variability at a province level, provides support for the hypothesis that the variability in *pfhrp2*-deletion frequency with transmission is driven by selection. However, in contrast to other reported surveys, the samples in this survey were primarily drawn from asymptomatic infections, and hence may not be representative of other reports of *pfhrp2*-deletion in symptomatic cases with higher parasite density. However, it is interesting to note that our results show broad agreement with published data sets from Zambia (*Laban et al., 2015*) and Ghana (*Amoah et al., 2016*) (*Table 1*). In particular, our predictions confirm that the HRP2 concern would be greater in Ghana than in Southern Zambia. However, one study in Senegal found a lower prevalence of *pfhrp2*-deletion than we predict (*Wurtz et al., 2013*).

A key uncertainty in predicting the potential spread of *pfhrp2*-deletion due to RDT-induced selective pressure is the extent of use of, and adherence to, RDT results and the availability of appropriate treatment. On the one hand, if adherence to RDT results is poor (for example, with patients who show continued clinical symptoms of malaria in the absence of a positive test) or additional microscopy-based detection is used (*Figure 1—figure supplement 3*), if appropriate treatment is not available (for example, due to stock-outs), or if treatment is not fully curative (for example, due to patient non-adherence, drug resistance or fake drugs) then the spread of these deletions will be slower than predicted. On the other hand, in areas in which active case detection occurs, or in which treatment is sought for non-malaria fevers (*Figure 1—figure supplement 4*), RDT-based treatment may also selectively clear non-deleted asymptomatic infections and hence increase the rate of spread of the deletion. However, when these factors, along with potential fitness costs associated with *pfhrp2*-deletion, were investigated together we still observed an increase in *pfhrp2*-deletion (*Figure 1—figure supplement 5*), which showed a similar rate of increase to that predicted by our model. Further data on RDT usage and adherence, as well as on non-malarial fevers and the precise fitness cost of *pfhrp2*-deletion, however, could help to refine mapping of areas of HRP2 concern.

A second uncertainty relates to the underlying prevalence of the mutation prior to RDT introduction. There is considerable variability in the estimates that have been measured both before and after RDT introduction, and it is entirely plausible that the presence of mutations could vary geographically at a range of spatial scales. However, estimating this variation is difficult given the lack

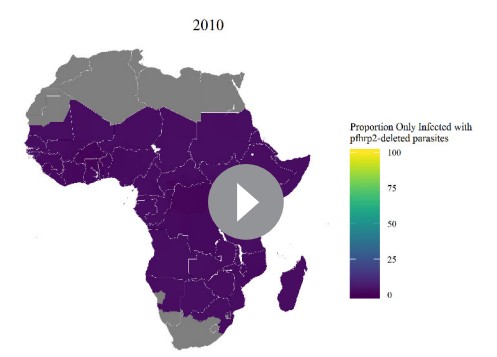

**Video 1.** The projected increase in individuals who are only infected with *pfhrp2*-deleted parasites, from 2010 to 2030, with an assumed starting frequency of 6% *pfhrp2*-deletion, and an assumed PfHRP3 epitope effect equal to 0.25%. The video relates directly to *Figure 4*.

DOI: https://doi.org/10.7554/eLife.25008.025

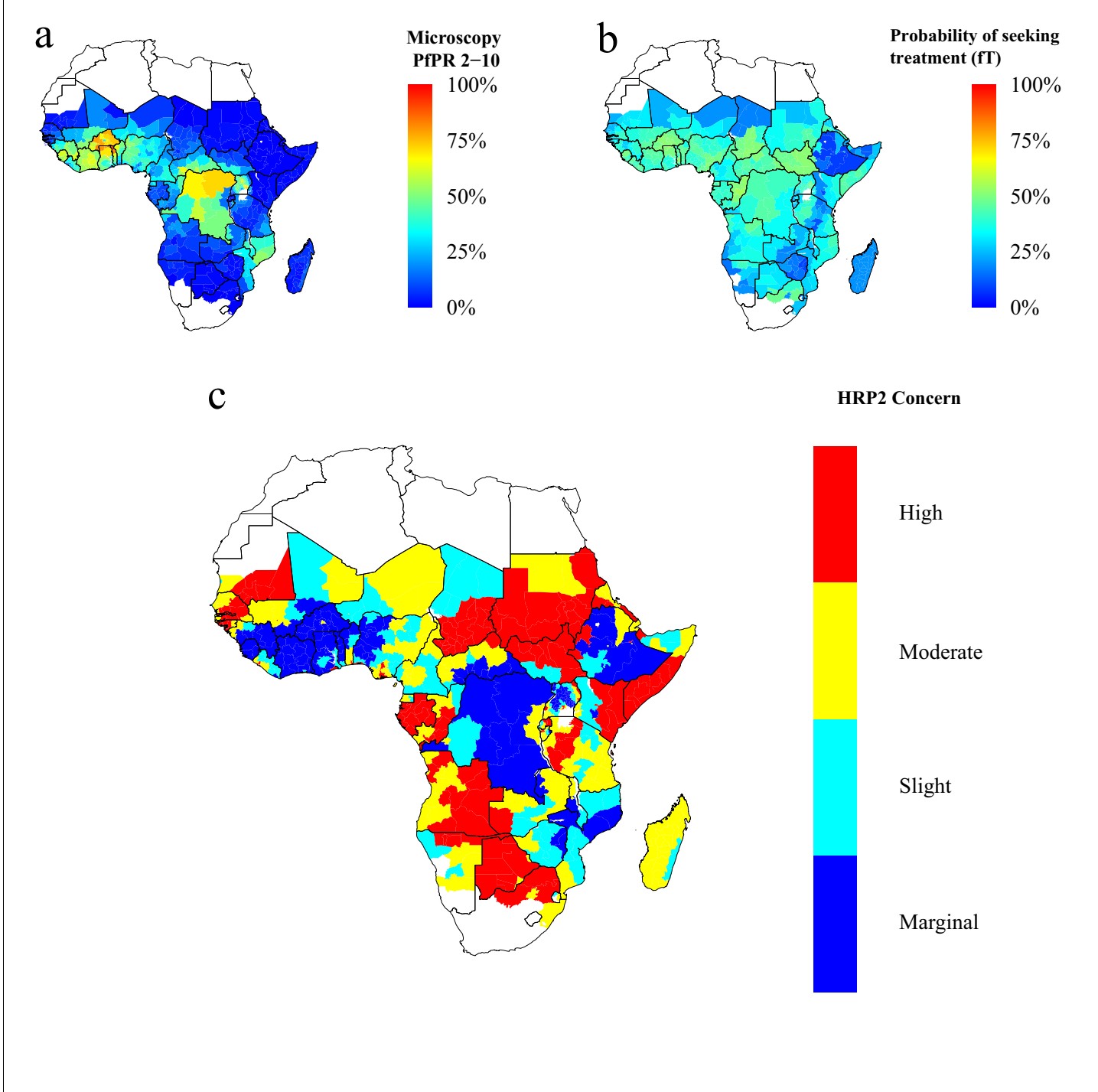

**Figure 4.** Predicted areas of HRP2 concern in comparison to recorded prevalence and treatment seeking rate, with an assumed probability of a clinical case seeking treatment, who is only infected with *pfhrp2*-deleted mutants, producing a positive RDT result ($\varepsilon$) equal to 0.25. The graphs show (a) the recorded malaria prevalence in children aged 2–10 by microscopy in 2010, (b) the frequency of people seeking treatment in 2010 and (c) the predicted concern for the impact of *pfhrp2*-deleted mutants. In (c), high, moderate and slight risk represent >20% infection due to only *pfhrp2*-deleted mutants by 2016, 2022 and 2030 respectively, and marginal risk represents <20% by 2030. In 2010, each region was assumed to have a starting frequency of 6% *pfhrp2*-deletion. Source data for **Figure 4** is provided within **Figure 4—source data 1**.

DOI: https://doi.org/10.7554/eLife.25008.018

The following source data and figure supplements are available for figure 4:

*Figure 4 continued on next page*

*Figure 4 continued*

**Source data 1.** Recorded malaria prevalence in children aged 2–10 by microscopy in 2010 (sourced from the Malaria Atlas mapping project [see Metadata - Datasets]), the frequency of people seeking treatment in 2010 (sourced from *Cohen et al., 2012* [see Metadata – Datasets]) and the simulated predicted concern for the impact of *pfhrp2*-deleted mutants, with an assumed PfHRP3 epitope effect equal to 0.25%.

DOI: https://doi.org/10.7554/eLife.25008.024

**Figure supplement 1.** Model malaria prevalence output against Malaria Atlas Project prevalence 2010 (*Bhatt et al., 2015*).

DOI: https://doi.org/10.7554/eLife.25008.019

**Figure supplement 2.** HRP2 Concern heat maps.

DOI: https://doi.org/10.7554/eLife.25008.020

**Figure supplement 3.** Predicted areas of HRP2 concern in comparison to recorded prevalence and treatment coverage with an assumed probability of a clinical case seeking treatment, who is only infected with pfhrp2-deleted mutants, producing a positive RDT result ($\varepsilon$) equal to 0.

DOI: https://doi.org/10.7554/eLife.25008.021

**Figure supplement 4.** Impact of different assumptions about starting frequency of *pfhrp2*-deletion upon the geographical pattern of selection-driven increase in *pfhp2*-deletion.

DOI: https://doi.org/10.7554/eLife.25008.022

**Figure supplement 5.** Years in which RDTs were used at community level in Sub-Saharan Africa.

DOI: https://doi.org/10.7554/eLife.25008.023

of a sampling framework in reports mainly based on clinical cases and given the relatively small sample sizes. Thus our results should be interpreted not as predictions of the absolute levels of the gene deletion, but rather indicative of geographical areas in which surveillance should be focused. Similarly, these results should not be interpreted as predictions of the precise negative impact on malaria prevalence as a result of increased gene deletions (*Figure 1—figure supplement 1*), but illustrative of the potential impact of false-negative test results upon malaria prevalence and the importance of alternative diagnostic methods (*Figure 1—figure supplement 3*). At the same time, further data collated in the coming months and years can be incorporated to iteratively update and refine our projections.

As with any modelling exercise, there are a number of important limitations. Firstly, we did not capture seasonality or any fitness cost associated with *pfhrp2*-deletion. At a given transmission level, highly seasonal locations are likely to have a lower frequency of *pfhrp2*-deletion in comparison to regions with perennial transmission. Seasonality could however cause substantial bottlenecks which may result in repetitive founder effects that could affect selection, resulting in either a decreased chance of *pfhrp2*-fixation or an accelerated fixation if it occurred (*Aguilée et al., 2009*). In simulations incorporating a fitness cost the selection pressure was found to be considerably weaker (*Figure 1—figure supplement 2*). The exact fitness cost, despite being unknown, is likely subtle as our modelled fitness penalty would cause the strain to be eliminated at less than 90% comparative fitness. In addition, current theories concerning the role of PfHRP2 indicate a more minor role in heme detoxification than previously thought. Strains lacking PfHRP2 have been shown to be viable (*Papalexis et al., 2001*), with heme detoxification more dependent on the recently characterised haem detoxification protein (HDP) (*Jani et al., 2008*). Furthermore, in South America the first cases of *pfhrp2*-deleted *P. falciparum* were confirmed prior to the introduction of RDTs (*Gamboa et al., 2010*). This suggests that these mutants may possess sufficiently high fitness such that the frequency of *pfhrp2*-deletion is maintained in the absence of a selective advantage exerted through the use of HRP2-based RDTs.

Secondly, our results depend on assumptions made regarding the contribution of PfHRP3 epitope cross-reactivity and the potential for false-positive RDT results. We found that increased cross-reactivity with PfHRP3 epitopes decreases selection for *pfhrp2*-deletion and was investigated due to confirmed observations of PfHRP2-based RDTs detecting PfHRP3 epitopes at high parasitaemia (*Baker et al., 2010*). In simulations with no epitope effect, the model predicts the pattern in DRC well (*Figure 3—figure supplement 1*) and predicts a higher overall estimate of HRP2 concern (*Figure 4—figure supplement 1*), although the same regional patterns are identified (*Figure 4—figure supplement 2*). Furthermore, false-positive RDT results would decrease the strength of the selection pressure, with *pfhrp2*-monoclonal infections being treated. However, false-positivity rates observed within round 6 of WHO RDT product testing were found to be low, with the median false-positive rate on both clean negative samples and samples containing other infectious agents equal to 0%,

and the overall false-positive rate on samples containing immunological factors equal to 0.9% (*World Health Organization, 2015c*).

Thirdly, in the absence of systematic country introduction data, we assumed introduction of RDTs in all countries from 2010 in accordance with the WHO recommendation of testing in 2010 (*World Health Organization, 2010*). The precise date from region to region is undoubtedly more complex, however 2010 is a sensible estimate given the reported years at which RDTs were available at the community level in SSA by the WHO (*Figure 4—figure supplement 5*) (*World Health Organization, 2012b*). However, the ratio of testing via microscopy versus RDT is likely to have decreased over this period, and hence our estimate of RDT use (which our model assumes is 100% from introduction) is likely too high. The sensitivity of the output to this parameter is demonstrated in the data from the DRC, in which higher levels of *pfhrp2*-deletion are observed in Kivu, an area in which RDT introduction likely occurred earlier than elsewhere in the country (*Médecins Sans Frontières, 2007*). Fourthly, extrapolating the starting frequency of *pfhrp2*-deletion strains from the DRC across the rest of SSA is a clear oversimplification; however, in the absence of similar datasets, we feel it provides a reasonable first estimate. To assess the implications of this estimate, we also considered how the pattern of geographical areas that we have recommended for priority surveillance changes under different assumed starting frequencies of *pfhrp2*-deletion (*Figure 4—figure supplement 4*). Despite the expected changes in the final frequency of *pfhrp2*-deletion in these settings, the overall pattern of areas with the highest selection-driven increase in *pfhrp2*-deletion remains the same. A final limitation is that we assumed that treatment rates and transmission of malaria remain constant from 2010. This is clearly not the case, with 30 countries in SSA reporting a decline in prevalence from 2010 to 2015 (*Bhatt et al., 2015*). These combined effects, however, would presumably cause an increase in monoclonal infections and subsequent false-negative RDTs due to *pfhrp2*-deleted parasites.

In summary, our modelling predicts that an increased emergence of *pfhrp2*-deleted mutants may be explained by the introduction of testing by PfHRP2-based RDTs. If this is indeed the case, this would be, to our knowledge, one of the first examples of the emergence of resistance of a pathogen to a diagnostic test. The use of these RDTs will result in the greatest selection pressure in regions that have low malaria transmission and a high frequency of RDT-based treatment of clinical cases. Rapid and accurate diagnosis of *P. falciparum* infection, however, is essential for continued reduction in malaria transmission. In light of this, it may be sensible for public health bodies who are responding to reports of *pfhrp2* gene deletions to focus surveillance in the regions we have identified as having a high HRP2 concern. This work should proceed alongside further improvement of non-HRP2-based RDTs, such as those that detect lactate dehydrogenase, and the development of new alternative diagnostics.

## Materials and methods

### *P. falciparum* transmission model

An individual-level stochastic model was developed to simulate the transmission dynamics of *Plasmodium falciparum*. The model is based upon previous modelling efforts (*Griffin et al., 2016*; *Griffin et al., 2014*; *Griffin et al., 2015*), and is described in full here before describing the extensions made with regards to PfHRP2 dynamics, and defining the parameters used and their sources. The model is implemented as stochastic individual-based model with a fixed daily time step, incorporating the necessary delay terms where mentioned, which is described in greater detail later. In overview, the transmission model considers people to exist in one of six infection states (*Figure 5*): susceptible (S), clinical disease (D), clinically diseased and receiving treatment (T), asymptomatic infection (A), protective state of prophylaxis (P), and subpatent infection (U).

Individuals begin life susceptible to infection (state S). At birth, individuals possess a level of maternal immunity that decays exponentially over the first 6 months. Each day individual $i$ is probabilistically exposed to infectious bites governed by their individual force of infection ($\Lambda_i$). $\Lambda_i$ is dependent on their pre-erythrocytic immunity, biting rate (dependent on both their age and their individual relative biting rate due to heterogeneous biting patterns in mosquitoes) and the mosquito population's size and infectivity. Infected individuals, after a latent period of 12 days ($d_E$), develop either clinical disease (state D) or asymptomatic infection (state A). This outcome is determined by

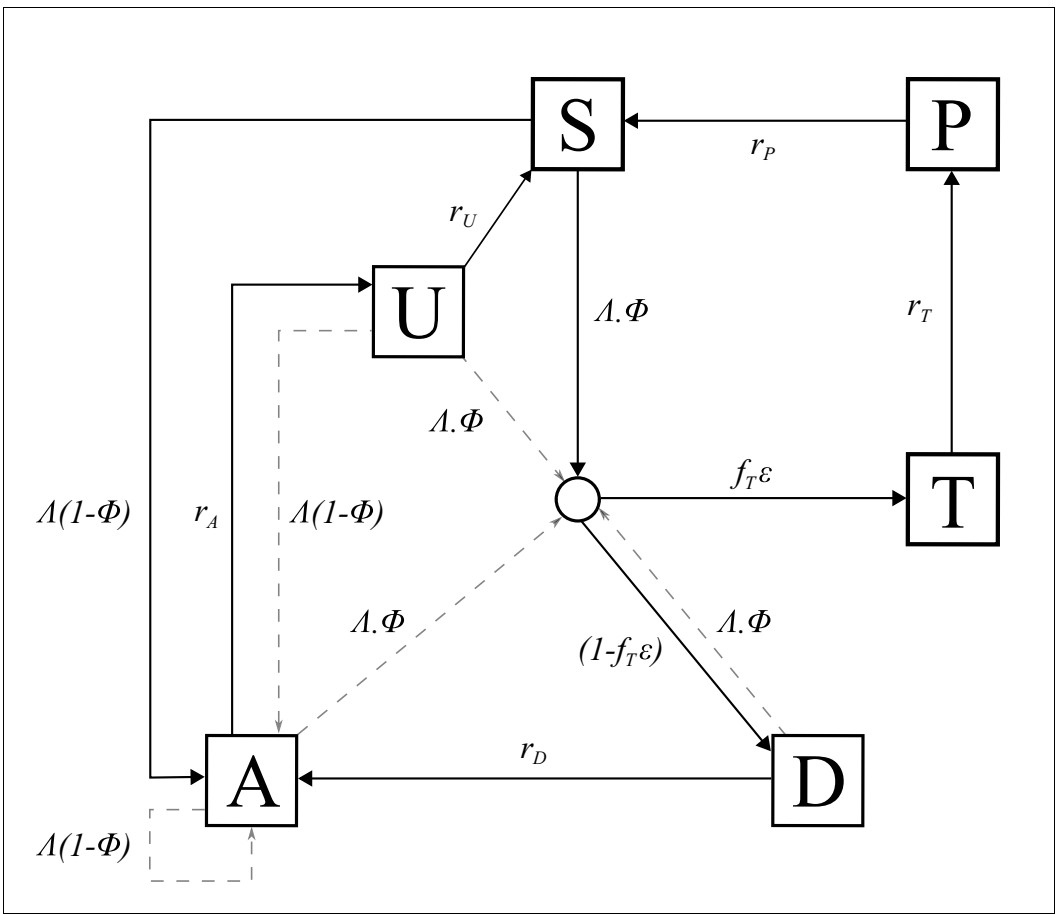

**Figure 5.** Transmission Model. Flow diagram for the human component of the transmission model, with dashed arrows indicating superinfection. S, susceptible; T, treated clinical disease; D, untreated clinical disease; P, prophylaxis; A, asymptomatic patent infection; U, asymptomatic sub-patent infection. All parameters are described within *Table 2*.

DOI: https://doi.org/10.7554/eLife.25008.026

their probability of acquiring clinical disease ($\phi$), which is dependent on their clinical immunity. Individuals that develop disease have a fixed probability ($f_T$) of seeking treatment (state T), and a variable probability ($\varepsilon_i$) that the clinical case yields a positive diagnostic result and subsequently receives treatment. $\varepsilon_i$ is dependent on the assumed role of PfHRP3 epitopes, and the strain profile of infected individual $i$ with respect to *pfhrp2*-deleted mutants. Treated individuals are assumed to always recover, i.e. fully-curative treatment, and then enter a protective state of prophylaxis (state P) at rate $r_T$, before returning to susceptible at rate $r_S$. Individuals that did not receive treatment recover to a state of asymptomatic infection at rate $r_D$. Asymptomatic individuals progress to a sub-patent infection (stage U) at rate $r_A$, before clearing infection and returning to susceptible at rate $r_U$. Additionally, superinfection is possible for all individuals in states D, A and U. Superinfected individuals who receive treatment will move to state T. Individuals who are superinfected but do not receive treatment in response to the superinfection will either develop clinical disease, thus moving to state D, or develop an asymptomatic infection and move to state A (except for individuals who were previously in state D, who will remain in state D).

The introduction of a fixed time step translates the waiting times at which individuals move from one infection state to another into a daily probability that this event occurs, with the probability drawn from the related exponential distribution. Thus the probability of a transition from state A to state B with hazard rate λ is given by:

$$Prob(A \rightarrow B): 1 - exp^{(-\lambda)}$$

The set of state transitions for individuals and their associated hazard rates are given below.

| Process | Transition | Hazard |
|---|---|---|
| Infection | $S \rightarrow I$ | $\Lambda_i(t - d_E)$ |
| Progression of untreated disease to asymptomatic infection | $D \rightarrow A$ | $r_D = \frac{1}{d_D}$ |
| Progression of asymptomatic infection to subpatent infection | $A \rightarrow U$ | $r_A = \frac{1}{d_A}$ |
| Progression of subpatent infection to susceptible | $U \rightarrow S$ | $r_U = \frac{1}{d_U}$ |
| Progression of treated disease to uninfected prophylactic period | $T \rightarrow P$ | $r_T = \frac{1}{d_T}$ |
| Progression from uninfected prophylactic period to susceptible | $P \rightarrow S$ | $r_P = \frac{1}{d_P}$ |
| Super-infection from untreated clinical disease, asymptomatic infection or subpatent infection | $D \rightarrow I$ $A \rightarrow I$ $U \rightarrow I$ | $\Lambda_i(t - d_E)$ |

Here state $I$ denotes an infection state which is not tracked but which leads to either clinical disease ($D$), treated clinical disease ($T$) or asymptomatic infection ($A$). In the original model the probability of entering these states is determined by drawing a sequence of Bernoulli trials for each infected individual as:

$$Prob(Clinical\ Disease): Bernoulli(\phi_i)$$

$$Prob(Treated\ Clinical\ Disease \mid Clinical\ Disease): Bernoulli(f_T)$$

For our model here, in which treatment is guided by RDT-based diagnostics, we introduce…

$$Prob(Treated\ Clinical\ Disease \mid Clinical\ Disease): Bernoulli(f_T \varepsilon_i)$$

We assume that each person has a unique biting rate, which is the product of their relative age dependent biting rate, $\psi_i$, given by

$$\psi_i(a) = \frac{\sum_{i=1}^{n} \psi_i(a)}{n} \left(1 - \rho \exp^{\frac{-a}{a_0}}\right)$$

and an assumed heterogeneity in biting patterns of mosquitoes, $\zeta_i$, which we assume persists throughout their lifetime and is drawn from a log-normal distribution with a mean of 1,

$$log(\zeta_i) \sim N\left(\frac{-\sigma^2}{2}, \sigma^2\right)$$

where $1 - \rho$ is the relative biting rate at birth when compared to adults and $a_0$ represents the time-scale at which the biting rate increases with age. The product of these biting rates is subsequently used to calculate an individual's entomological inoculation rate, $h_i$, and subsequently their force of infection, which are given by

$$h_i = \alpha_k I_M \zeta_i \psi_i$$

$$\Lambda_i = h_i b_i$$

where $\alpha_k$ is the daily rate at which a mosquito takes a blood meal, $I_M$ is the size of the infected mosquito population, and $b_i$ is the probability of infection given an infectious mosquito bite.

The human population was assumed to have a maximum possible age of 100 years, with an average age of 21 years within the population yielding an approximately exponential age distribution typical of sub-Saharan countries. When an individual dies, they are replaced with a new-born individual whose individual biting rate due to heterogeneity in biting patterns is drawn again from a log-normal distribution with a mean of 1.

**Table 2.** Parameters used within the human transmission and mosquito population models.

| Parameter | Symbol | Estimate |
|---|---|---|
| **Human infection duration (days)** | | |
| Latent period | $d_E$ | 12 |
| Patent infection | $d_A$ | 200 |
| Clinical disease (treated) | $d_T$ | 5 |
| Clinical disease (untreated) | $d_D$ | 5 |
| Sub-patent infection | $d_U$ | 110 |
| Prophylaxis following treatment | $d_P$ | 25 |
| **Treatment Parameters** | | |
| Probability of seeking treatment if clinically diseased | $f_T$ | Variable |
| Probability of a clinical case seeking treatment, who is only infected with *pfhrp2*-deleted mutants, producing a positive RDT result. | $\varepsilon$ | 0 or 0.25 |
| **Infectiousness to mosquitoes** | | |
| Lag from parasites to infectious gametocytes | $d_g$ | 12 days |
| Untreated disease | $c_D$ | 0.0680 day$^{-1}$ |
| Treated disease | $c_T$ | 0.0219 day$^{-1}$ |
| Sub-patent infection | $c_U$ | 0.000620 day$^{-1}$ |
| Parameter for infectiousness of state A | $\gamma_1$ | 1.824 |
| **Age and heterogeneity** | | |
| Age-dependent biting parameter | $\rho$ | 0.85 |
| Age-dependent biting parameter | $a_0$ | 8 years |
| Daily mortality rate of humans | $\mu$ | 0.000130 |
| Variance of the log heterogeneity in biting rates | $\sigma^2$ | 1.67 |
| **Immunity reducing probability of infection** | | |
| Maximum probability due to no immunity | $b_0$ | 0.590 |
| Maximum relative reduction due to immunity | $b_1$ | 0.5 |
| Inverse of decay rate | $d_B$ | 10 years |
| Scale parameter | $I_{B0}$ | 43.879 |
| Shape parameter | $\kappa_B$ | 2.155 |
| Duration in which immunity is not boosted | $u_B$ | 7.199 |
| **Immunity reducing probability of clinical disease** | | |
| Maximum probability due to no immunity | $\phi_0$ | 0.791 |
| Maximum relative reduction due to immunity | $\phi_1$ | 0.000737 |
| Inverse of decay rate | $d_{CA}$ | 30 years |
| Scale parameter | $I_{C0}$ | 18.0237 |
| Shape parameter | $\kappa_C$ | 2.370 |
| Duration in which immunity is not boosted | $u_C$ | 6.0635 |
| New-born immunity relative to mother's | $P_M$ | 0.774 |
| Inverse of decay rate of maternal immunity | $d_M$ | 67.695 |
| **Immunity reducing probability of detection** | | |
| Minimum probability due to maximum immunity | $d_1$ | 0.161 |
| Inverse of decay rate | $d_{ID}$ | 10 years |
| Scale parameter | $I_{D0}$ | 1.578 |
| Shape parameter | $\kappa_D$ | 0.477 |
| Duration in which immunity is not boosted | $u_D$ | 9.445 |
| Scale parameter relating age to immunity | $a_D$ | 21.9 years |

*Table 2 continued on next page*

*Table 2 continued*

| Parameter | Symbol | Estimate |
|---|---|---|
| Time-scale at which immunity changes with age | $f_{D0}$ | 0.00706 |
| Shape parameter relating age to immunity | $\gamma_D$ | 4.818 |
| PCR detection probability parameters state A | $\alpha_A$ | 0.757 |
| PCR detection probability parameters state U | $\alpha_U$ | 0.186 |
| Mosquito Population Model | | |
| Daily mortality of adults | $\mu_M$ | 0.132 |
| Daily biting rate | $\alpha_k$ | 0.307 |
| Extrinsic incubation period | $d_{EM}$ | 10 days |

DOI: https://doi.org/10.7554/eLife.25008.028

## Immunity and detection functions

We consider three stages at which immunity may impact transmission:

1. Pre-erythrocytic immunity, $I_B$; reduction in the probability of infection given an infectious mosquito bite.
2. Acquired and Maternal Clinical Immunity, $I_{CA}$ and $I_{CM}$ respectively; reduction in the probability of clinical disease given an infection due to the effects of blood stage immunity.
3. Detection immunity, $I_D$; reduction in the probability of detection and a reduction in the onward infectivity towards mosquitoes due to the effects of blood stage immunity.

Maternal clinical immunity is assumed to be at birth a proportion, $P_M$, of the acquired immunity of a 20 year-old and to decay at rate $\frac{1}{d_M}$.

The probabilities of infection, detection and clinical disease are subsequently created by transforming each immunity function by Hill functions. An individual's probability of infection, $b_i$, is given by

$$b_i = b_0 \left( b_1 + \frac{1 - b_1}{1 + \left(\frac{I_B}{I_{B0}}\right)^{\kappa_B}} \right)$$

where $b_0$ is the maximum probability due to no immunity, $b_0 b_1$ is the minimum probability and $I_{B0}$ and $\kappa_B$ are scale and shape parameters respectively.

An individual's probability of clinical disease, $\phi_i$, is given by

$$\phi_i = \phi_0 \left( \phi_1 + \frac{1 - \phi_1}{1 + \left(\frac{I_{CA} + I_{CM}}{I_{C0}}\right)^{\kappa_C}} \right)$$

where $\phi_0$ is the maximum probability due to no immunity, $\phi_1 \phi_0$ is the minimum probability and $I_{C0}$ and $\kappa_C$ are scale and shape parameters respectively.

An individual's probability of being detected by microscopy when asymptomatic, $q_i$, is given by

$$q_i = d_1 + \left( \frac{1 - d_1}{1 + \left(\frac{I_D}{I_{D0}}\right)^{\kappa_D} f_D} \right)$$

where $d_1$ is the minimum probability due to maximum immunity, and $I_{D0}$ and $\kappa_D$ are scale and shape parameters respectively. $f_D$ is dependent only on an individual's age is given by

$$\frac{df_D}{da} = 1 - \frac{1 - f_{D0}}{1 + \left(\frac{a}{a_D}\right)^{\gamma_D}}$$

where $f_{D0}$ represents the time-scale at which immunity changes with age, and $a_D$ and $\gamma_D$ are scale

and shape parameters respectively. Lastly, $\alpha_A$ and $\alpha_U$ are parameters that determine the probability that an individual in states A and U are detectable by PCR, which are given by $q^{\alpha_A}$ and $q^{\alpha_U}$ respectively.

The contribution made by each infected individual towards the overall infectiousness of the human population towards mosquitoes is proportional to both their infectious state and their probability of detection, with a lower probability of detection assumed to correlate with a lower parasite density. Individuals who are in state D (clinically diseased), state U (sub-patent infection) and state T (receiving treatment) contribute $c_D$, $c_U$ and $c_T$. In state A, infectious contribution, $c_A$, is given by $c_U + (c_D - C_U)q^{\gamma_I}$ where $q$ is the probability of being detected by microscopy when asymptomatic, and $\gamma_I$ is a parameter that controls how quickly infectiousness falls within the asymptomatic state.

## Stochastic model equations

Given the definitions above, the full stochastic individual-based human component of the model can be formally described by its Kolmogorov forward equations. As before, let $i$ index individuals in the population. Then the state of individual $i$ at time $t$ is given by $\{j, k, t_k, l, t_l, m, t_m, cm, a, t\}$, where $a$ is age, $j$ represents infection status ($S, D, A, U, T$ or $P$), $k$ is the level of infection-blocking immunity and $t_k$ is the time at which infection blocking immunity was last boosted. Similarly, $l$ and $t_l$ denote the level and time of last boosting of clinical immunity, respectively, while $m$ and $t_m$ do likewise for parasite detection immunity, and $cm$ represents maternal immunity. Let $\delta_{p,q}$ denote the Kronecker delta ($\delta_{p,q} = 1$ if $p = q$ and 0 otherwise) and $\delta(x)$ denote the Dirac delta function. Defining $P_i(j, k, t_k, l, t_l, m, t_m, cm, a, t)$ as the probability density function for individual $i$ being in state $\{j, k, t_k, l, t_l, m, t_m, cm, a, t\}$ at time $t$, the time evolution of the system is governed by the following forward equation:

$$\frac{\partial P_i(j, k, t_k, l, t_l, m, t_m, cm, a, t)}{\partial t} + \frac{\partial P_i(j, k, t_k, l, t_l, m, t_m, cm, a, t)}{\partial a} = \tag{1}$$

$$\delta_{j,S}[r_P P_i(P, k, t_k, l, t_l, m, t_m, cm, a, t) + r_U P_i(U, k, t_k, l, t_l, m, t_m, cm, a, t)] \tag{2}$$

$$+\delta_{j,A}[r_D P_i(D, k, t_k, l, t_l, m, t_m, cm, a, t)] \tag{3}$$

$$+\delta_{j,U}[r_A P_i(A, k, t_k, l, t_l, m, t_m, cm, a, t)] \tag{4}$$

$$+\delta_{j,P}[r_T P_i(T, k, t_k, l, t_l, m, t_m, cm, a, t)] \tag{5}$$

$$+(1 - b_i)h_i(t - d_E)[\delta_{j,S} + \delta_{j,D} + \delta_{j,A} + \delta_{j,U}]O_b \diamond P_i(j, k, t_k, l, t_l, m, t_m, cm, a, t) \tag{6}$$

$$+b_i h_i(t - d_E) \left[ \delta_{j,A}(1 - \phi_i) + \delta_{j,D}\phi_i(1 - \varepsilon_i f_T) + \delta_{j,T}\phi_i \varepsilon_i f_T \right] \\ O_b \diamond O_c \diamond O_b \diamond \sum_{j' \in \{S,A,U\}} P_i\left(j', k, t_k, l, t_l, m, t_m, cm, a, t\right) \tag{7}$$

$$+b_i h_i(t - d_E) \left[ +\delta_{j,D}\phi_i(1 - \varepsilon_i f_T) + \delta_{j,T}\phi_i \varepsilon_i f_T \right] \\ O_b \diamond O_c \diamond O_d \diamond P_i\left(D, k, t_k, l, t, m, t_m, cm, a, t\right) \tag{8}$$

$$+\left[ r_B k \frac{\partial}{\partial k} + r_{CA} l \frac{\partial}{\partial l} + r_{ID} m \frac{\partial}{\partial m} + r_{CM} cm \frac{\partial}{\partial cm} \right] P_i(j, k, t_k, l, t_l, m, t_m, cm, a, t) \tag{9}$$

$$+\mu \, \delta(a)\delta\left(t_k + T_{big}\right)\delta\left(t_l + T_{big}\right)\delta\left(t_m + T_{big}\right)\delta_{j,S}\delta_{k,0}\delta_{l,0}\delta_{m,0} \sum_{j'} P_i\left(j', k, t_k, l, t_l, m, t_m, cm, a, t\right) \tag{10}$$

$$-[\mu + r_P \delta_{j,P} + r_U \delta_{j,U} + r_D \delta_{j,D} + r_A \delta_{j,A} + r_T \delta_{j,P} \\ +h_i(t - d_E)[\delta_{j,S} + \delta_{j,D} + \delta_{j,A} + \delta_{j,U}]]P_i(j, k, t_k, l, t, m, t_m, cm, a, t) \tag{11}$$

Here $O_b$, $O_c$ and $O_d$ are commutative integral operators with the following action on an arbitrary density $f(j, k, t_k, l, t_l, m, t_m, cm, a, t)$:

$$O_b \diamond f = \delta(t - t_k) \int_0^\infty f(j, k-1, t-u_B-\tau, l, t_l, m, t_m, cm, a, t)d\tau + \theta\left(\frac{t-t_k}{u_B}\right)f(j, k, t_k, l, t_l, m, t_m, cm, a, t)$$

$$O_c \diamond f = \delta(t - t_l) \int_0^\infty f(j, k, t_k, l-1, t-u_C-\tau, m, t_m, cm, a, t)d\tau + \theta\left(\frac{t-t_l}{u_C}\right)f(j, k, t_k, l, t_l, m, t_m, cm, a, t)$$

$$O_d \diamond f = \delta(t - t_m) \int_0^\infty f(j, k, t_k, l, t_l, m-1, t-u_D-\tau, cm, a, t)d\tau + \theta\left(\frac{t-t_m}{u_D}\right)f(j, k, t_k, l, t_l, m, t_m, cm, a, t).$$

Finally, $\theta(x)$ is an indicator function such that $\theta(x) = 1$ if $x<1$ and 0 otherwise. These functions allow the fixed periods of time in which immunities are not boosted after a previous boost to be included within the stochastic equations, while also allowing superinfection events to be incorporated.

For simulation, a discrete time approximation of this stochastic model was used, with a time-step of 1 day. For each individual $k$, $l$ and $m$ are set to zero at birth, while $t_k$, $t_l$ and $t_m$ are set to a large negative value $-T_{big}$ (to represent never having been exposed or infected). Each immunity term increases by 1 for an individual whenever that individual receives an infectious bite ($k$), or is infected ($l$ and $m$), if the previous boost to $k$, $l$ and $m$ occurred more than $u_B$, $u_C$ and $u_D$ days earlier, respectively. Immunity levels decay exponentially at rate $r_B$, $r_{CA}$ and $r_{ID}$, where $r_B$, $r_{CA}$ and $r_{ID}$ are equal to $\frac{1}{d_B}, \frac{1}{d_{CA}}$ and $\frac{1}{d_{ID}}$ respectively.

The stochastic model equations detailed above can be explained as follows. The first line is the total time derivative of $P_i(j, k, t_k, l, t_l, m, t_m, cm, a, t)$. The next four lines describe the flows into states S, A, U and P due to progression through infection states.

The sixth line describes exposure to malaria that boosts pre-erythrocytic immunity but does not lead to an infection. The first term within the commutative integral operator $O_b$ here considers the density of individuals who are in immunity class $k$-1 and whose last boost to their pre-erythrocytic immunity was more than $u_B$ days earlier, and thus will be flowing into the considered density, $k$, from a lower pre-erythrocytic immunity. The second term in the integral will equate to 1 when considering individuals who are in immunity class $k$ and whose last boost to their pre-erythrocytic immunity was less than $u_B$ days earlier and thus do not see their immunity boosted and hence remain in class $k$. This is needed to represent the current density of individuals in the considered density. There is no term for individuals in immunity class $k$ whose last boost to their pre-erythrocytic immunity occurred more than $u_B$ days earlier as they would move out of the considered density (into class $k$ + 1) and hence the indicator function will equate to 0 for these individuals.

The seventh line describes exposure events occurring to individuals in states S, A and U which do result in patent (blood-stage) infection, resulting in transition into states A, D or T. The force of infection acting on the density in state D is not included here but rather in the eighth line since these individuals may only move to states T or D and not A. In both the seventh and eighth lines, the commutative operators here function as described earlier. This tracks the density of individuals in immunity states one lower whose last boost to any of the three immunity types occurred a sufficient number of days earlier to flow into the considered density, while also considering the individuals already at the same immunity as the considered density to remain in their current combined infection/immunity state (when the indicator function equates to 1) or to move to a new infection/immunity state (when the indicator function equates to 0).

The ninth line (effectively a first order wave equation) represents deterministic exponential decay of the four different types of immunity. The tenth line represents the birth process. We assume a constant population size, so upon death, individuals flow into the state with no immunity and last immunity boosting times are set to $-T_{big}$, chosen to be sufficiently early to allow immediate boosting upon exposure to infection (i.e. zero immunity other than maternal at birth). The last line shows the removal of individuals from the population through death, balancing the inflow from the previous line.

## Mosquito population dynamics

The adult stage of mosquito development was modelled in a compartmental formulation. Susceptible adult mosquitoes ($S_M$) become infected at a rate which is proportional to the infectiousness of the human population lagged by $d_g$ days, which represents the delay from emergence of asexual blood-stage parasites to sexual gametocytes that contribute towards onward infectivity. The force of infection towards mosquitoes on a given day, $\Lambda_M$, is represented by the sum of the contributions from each infected human, delayed by $d_g$, towards the overall infectiousness of the human population, which is given by

$$\Lambda_M = \frac{\alpha_k}{N} \left( \sum_{i=1}^{\Sigma_D} \zeta_i \psi_i c_D + \sum_{i=1}^{\Sigma_T} \zeta_i \psi_i c_T + \sum_{i=1}^{\Sigma_A} \zeta_i \psi_i c_A + \sum_{i=1}^{\Sigma_U} \zeta_i \psi_i c_U \right) (t - d_g)$$

Infected mosquitoes then pass through a latent stage ($E_M$) of duration $d_{EM}$, before becoming infectious to humans ($I_M$). Infectious mosquitoes remain infectious until they die. The differential equations governing the adult stage of mosquitoes are given by

$$\frac{dS_M}{dt} = \mu_M M_v - \mu_M S_M - \Lambda_M S_M$$

$$\frac{dE_M}{dt} = \Lambda_M S_M - \mu_M E_M - \Lambda_M (t - d_{EM}) S_M (t - d_{EM}) exp^{-\mu_M d_{EM}}$$

$$\frac{dI_M}{dt} = \Lambda_M (t - d_{EM}) S_M (t - d_{EM}) exp^{-\mu_M d_{EM}} - \mu_M I_M$$

where $\mu_M$ is the daily death rate of adult mosquitoes, and $M_v$ is the total mosquito population, i.e. $S_M + E_M + I_M$.

## PfHRP2 dynamics

Individuals that are newly infected receive either a *pfhrp2*-deleted mutant or a wild type, determined probabilistically by the ratio of *pfhrp2*-deleted mutants in the contribution to onwards infectiousness governed by the human infectious population delayed by $d_{EM}$. An individual with clinical disease (state D), who possesses an equal number of wild type and *pfhrp2*-deleted mutants will for example contribute ½$c_D$ to both the wild type and the mutant profile of the onwards infectiousness to mosquitos. In simulations incorporating a fitness cost associated with *pfhrp2*-deletion the contribution terms ($c_D$, $c_T$, $c_A$ and $c_U$) required to calculate the contribution to the human infectious reservoir made by the deletion strains are comparatively decreased relevant to the wild type strains in order to represent an assumed decrease in parasitaemia and onward transmission. This effect would also capture if the *pfhrp2*-deleted strain is comparatively less fit within the mosquito. This would be of importance when considering blood meals taken by mosquitoes feeding on polyclonally infected individuals, in which we would expect the fitter wild-type parasite to be probabilistically more likely to be onwardly transmitted.

If a newly infected individual is only infected with *pfhrp2*-deleted mutants the probability that they enter the treated class is $\varepsilon f_T$, where $\varepsilon$ is equal to the cross reactivity contribution of PfHRP3 epitopes. If, however, they contain any wild type strains, $\varepsilon$ is assumed to always equal 1. Additionally, if a subpatent individual is superinfected resulting in clinical disease, $\varepsilon$ is assumed to equal the cross reactivity contribution of PfHRP3 epitopes if the acquired strain from superinfection is *pfhrp2*-deleted. Analogously, $\varepsilon$ is assumed to always equal 1 if the acquired strain was wild type. This is to reflect the inability of RDTs to detect any of the strains that were previously present within the subpatent individual.

Individuals that clear infection lose all strains, and infected individuals clear a random strain at rate $nr_C$, where $n$ is the total number of strains and $r_C$ is the rate at which one strain is cleared in a monoinfected individual, that is $\frac{1}{d_A + d_U}$. This introduces a carrying capacity on the number of strains an individual can be infected with, which scales with the transmission intensity. The multiplicity of infection and strain profile of an infected individual have no effect on the disease outcome except when the use of RDTs is introduced.

## Model code availability and parameter values

The model code was developed using the R language (RRID:SCR_001905), (*R Core Team, 2016*) and is available with shape files and plotting scripts through an open source MIT license at https://github.com/OJWatson/hrp2malaRia (*Watson, 2017*). A copy is archived at https://github.com/elifesciences-publications/hrp2malaRia. The model is also written out in full as a pseudocode model using mathematical syntax (*Supplementary file 1*). Parameter estimates used within the model were taken from *Griffin et al. (2014)*, (*2015*) and (*Griffin et al., 2016*) however have been included in *Table 2* for clarity.

## Characterising the epidemiological and clinical drivers of selection for *pfhrp2*-deletion

The rate of *pfhrp2*-deleted mutant emergence after the introduction of RDTs was examined across a range of malaria transmission intensities (10%, 25% and 60% parasite prevalence across all ages [PfPR]) and starting proportions of *pfhrp2*-deleted mutants (2%, 8% and 12% mutants). For all simulations conducted, the proportion of clinically diseased cases seeking treatment was equal to 40% ($f_T$ = 0.4). In all simulations ten stochastic realisations of 100,000 individuals were simulated for 60 years to reach equilibrium first, before exploring different parameter settings. RDTs were then introduced and the proportion of strains that are *pfhrp2*-deleted recorded over the following 10 years. The effect of transmission intensities (0%–90% PfPR) was explored further by recording both the proportion of strains that are *pfhrp2*-deleted and the proportion of individuals only infected with *pfhrp2*-deleted mutants at 5 years after RDT introduction. In these simulations PfHRP3 epitopes were assumed to never yield a positive RDT result ($\varepsilon$ = 0.0).

The rate of emergence was further examined under different assumptions about the proportion of people seeking treatment ($f_T$ = 0.2–0.6), recording the time taken for the proportion of individuals only infected with *pfhrp2*-deleted mutants to reach 20%. For each simulation we assumed 8% of strains carried *pfhrp2*-deleted mutants prior to RDT introduction. We also considered the role that PfHRP3 antigens may have in the performance of PfHRP2-based RDTs, assuming that 25% of individuals only infected with *pfhrp2*-deleted mutants receive treatment due to the presence of PfHRP3 epitopes ($\varepsilon$ = 0.25), based on an estimate of PfHRP2-based RDT cross-reactivity (*Baker et al., 2005*).

A series of analyses were additionally conducted to characterise the impact of a number of assumptions within the model. These sensitivity analyses were conducted at 20% PCR PfPR across all ages, with the proportion of clinically diseased cases seeking treatment equal to 40% ($f_T$ = 0.4), and an assumed starting frequency of *pfhrp2*-deletion equal to 10%. As before, in all simulations ten stochastic realisations of 100,000 individuals were simulated for 60 years to reach equilibrium first, before exploring different parameter settings.

We initially assessed the impact upon the strength of selection of a range of assumed comparative fitness costs associated with *pfhrp2*-deletion (5%–100%). Secondly, we explored the impact on selection of introducing additional microscopy-based diagnosis, occurring in 30% of cases in alignment with the 71% use of RDTs in 2014 (*World Health Organization, 2015a*), with and without an assumed non-adherence to RDT results in 10% of cases in alignment with estimated improving levels of adherence to RDT results (*D'Acremont et al., 2013*). Lastly, we investigated the impact of non-malarial fevers (NMF), introducing an estimate for the annual NMF rate. This estimate was sourced by first finding household Demographic Health Surveys in SSA that surveyed whether individuals had been previously sick with a fever in the last 2 or 4 weeks and if and where they sought treatment for that fever. The resultant six surveys (*Institut de Statistiques et d'Études Économiques du Burundi - ISTEEBU et al., 2013*), (*Institut de Statistiques et d'Études Économiques du Burundi - ISTEEBU et al., 2013*) Liberia (2009 [*National Malaria Control Program - NMCP/Liberia, Ministry of Health and Social Welfare/Liberia et al., 2012*] and 2011 [*National Malaria Control Program - NMCP/Liberia, Ministry of Health and Social Welfare et al., 2009*]), Mali (2010) and Nigeria (2010 [*National Population Commission - NPC/Nigeria et al., 2012*] and 2015 [*National Malaria Elimination Programme - NMEP/Nigeria et al., 2016*]) were then subset by those that had sufficiently representative fever data across all ages, which yielded five surveys with Mali failing to be sufficiently representative at higher age ranges. The survey from Burundi was chosen for further analysis as both surveys from Liberia and Nigeria reported substantial treatment sought from drug peddlers and drug hawkers respectively, complicating inference on the clinical outcome of any treatment sought

**Table 3.** HRP2 classifiers used in sub-Saharan Africa mapping assuming RDT introduction in 2010.

| Proportion of population only infected with *pfhrp2*-deleted mutants | Concern classifier |
|---|---|
| >20% by 2016 | High |
| >20% by 2022 | Moderate |
| >20% by 2030 | Slight |
| <20% by 2030 | Marginal |

DOI: https://doi.org/10.7554/eLife.25008.030

for the fever. From this survey an age-bracketed annual rate of fever that led to treatment being sought was calculated, with smaller age brackets used at younger ages to capture the rapid change in fever rates at younger ages. This annual estimate was then scaled by 57% to represent the likely NMF rate, as estimated from a large scale estimate across Africa (*Gething et al., 2010*). We subsequently incorporated this rate to explore the impact of NMF upon selection. Within these simulations, we assume that individuals currently treated or in prophylaxis will not receive further antimalarial treatment when presenting with a NMF. Susceptible and subpatent individuals who seek treatment due to a NMF will only receive treatment due to non-adherence to test results. Lastly, asymptomatic and diseased individuals who seek treatment due to a NMF will always receive treatment, unless they are monoinfected with *pfhrp2*-deleted parasites in which case they will only be treated due to potential PfHRP3 epitope contributions, non-adherence to RDT results or if they were diagnosed with microscopy-based diagnosis.

## Estimating the starting frequency and geographic spread of *pfhrp2*-deletion

To estimate the current and future proportion of *pfhrp2*-deleted mutants across SSA, we require a starting frequency of *pfhrp2*-deletion. We used estimates of the proportion of *pfhrp2*-deleted mutants (*Parr et al., 2016*) from the 2013–2014 DRC Demographic and Health Survey (DHS) (*Meshnick et al., 2015*) to infer the starting frequency before RDTs were introduced in 2010–2011, (*Meshnick et al., 2015*) using the weighted PCR prevalence of malaria in children aged 6–59 months (PCR PfPR 6–59 months) and the reported frequency of people seeking treatment in the 26 Divisions Provinciales de la Santé (DPS). The DHS survey was a nationally representative cross-sectional study of 7137 children aged 6–59 months and 783 subjects with RDT-/PCR+ results were tested using PCR assays to detect and confirm *pfhrp2*-deletion.

We explored 50 starting frequencies between 0.1%–10%, with an assumed probability of a clinical case seeking treatment, who is only infected with *pfhrp2*-deleted mutants, producing a positive RDT result ($\varepsilon$) equal to 0.25. RDTs were assumed to be introduced in 2010 except for North- and South-Kivu where the use of RDTs occurred from 2007 in the refugee camps. (*Médecins Sans Frontières, 2007*; *United Nations High Commissioner for Refugees, 2013*) For each starting frequency, ten stochastic realisations of 100,000 individuals were simulated for each DPS at malaria prevalence levels aligned to the observed weighted PCR prevalence of malaria in children aged 6–59 months for these provinces. These simulations were run for 60 years prior to the introduction of RDTs to ensure equilibrium was reached. The output from each set of simulations at a given starting frequency was smoothed using a local regression (LOESS) model, and the starting frequency identified as the set of simulations with the smallest residual sum of squares when compared to the recorded relationship from the DHS survey. It is important to highlight that due to the non-spatial nature of the model, each geographical region simulated occurs independently to neighbouring regions, i.e. there is no spatial spread of parasites between regions. Additionally, novel mutation emergence was not modelled explicitly and thus stochastic loss of the *pfhrp2*-deletion genotype would always yield a final *pfhrp2*-deletion frequency of 0%.

The estimated starting frequency was then used to simulate trends in the prevalence of *pfhrp2*-deleted mutants across SSA, exploring a range of treatment coverages and transmission intensities, with $\varepsilon$ = 0.25. These simulations considered populations of 100,000 individuals that were simulated for 20 years from 2010 to 2030, with the introduction of RDTs assumed across all regions in 2010. These outputs were matched to the mean microscopy-based PfPR in 2–10 year olds (PfPR$_{2-10}$) in 2010 by first administrative unit and estimates of the proportion of cases seeking treatment from

previously modelled estimates using the DHS and the Malaria Indicator Cluster Surveys (*Cohen et al., 2012*). The time taken for the proportion of infections due to only *pfhrp2*-deleted mutants to reach 20% was recorded and classified to map areas of HRP2 concern under four qualitative classifications shown in *Table 3*.

## Additional information

### Funding

| Funder | Grant reference number | Author |
|---|---|---|
| Wellcome | 109312/Z/15/Z | Oliver John Watson |
| National Institute of Allergy and Infectious Diseases | 5R01AI107949 | Steven R Meshnick |
| Imperial College London | | Hannah C Slater |
| Medical Research Council | MR/N01507X/1 | Robert Verity |
| Department for International Development | | Azra C Ghani |
| National Institute of Allergy and Infectious Diseases | 5T32AI007151 | Jonathan B Parr |

The funders had no role in study design, data collection and interpretation, or the decision to submit the work for publication.

### Author contributions

Oliver J Watson, Conceptualization, Software, Formal analysis, Methodology, Writing—original draft, Writing—review and editing; Hannah C Slater, Robert Verity, Conceptualization, Formal analysis, Supervision, Methodology, Writing—review and editing; Jonathan B Parr, Conceptualization, Resources, Formal analysis, Supervision, Methodology, Writing—review and editing; Melchior K Mwandagalirwa, Steven R Meshnick, Resources, Formal analysis, Writing—review and editing; Antoinette Tshefu, Resources, Writing—review and editing; Azra C Ghani, Conceptualization, Resources, Formal analysis, Supervision, Methodology, Project administration, Writing—review and editing

### Author ORCIDs

Oliver J Watson http://orcid.org/0000-0003-2374-0741

### Decision letter and Author response

Decision letter https://doi.org/10.7554/eLife.25008.033
Author response https://doi.org/10.7554/eLife.25008.034

## Additional files

### Supplementary files

• Supplementary file 1: Simulation model pseudocode. Mathematical style pseudocode description of the simulation model.
DOI: https://doi.org/10.7554/eLife.25008.029

• Transparent reporting form
DOI: https://doi.org/10.7554/eLife.25008.031

### Major datasets

The following previously published dataset was used:

| Author(s) | Year | Dataset title | Dataset URL | Database, license, and accessibility information |
|---|---|---|---|---|
| Bhatt S, Weiss DJ, Cameron E, Bisanzio D, Mappin B, Dalrymple U, Battle KE, Moyes CL, Henry A, Eckhoff PA, Wenger EA, Briët O, Penny MA, Smith TA, Bennett A, Yukich J, Eisele TP, Griffin JT, Fergus CA, Lynch M, Lindgren F, Cohen JM, Murray CLJ, Smith DL, Hay SI, Cibulskis RE, Gething PW | 2015 | PfPR2-10 in Africa 2000-2015 | http://www.map.ox.ac.uk/static/africa-now/data_downloads/prevalence/tables/admin1/PfPR_population_weighted.zip | Publicly available at the Malaria Atlas Project under a Creative Commons license (http://www.map.ox.ac.uk/) |

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
