## [Decision Letter]

Thank you for submitting your article "Modelling the drivers of the spread of *Plasmodium falciparum hrp2* gene deletions in sub-Saharan Africa" for consideration by *eLife*. Your article has been reviewed by three peer reviewers, one of whom, Ben Cooper (Reviewer #1), is a member of our Board of Reviewing Editors, and the evaluation has been overseen by Prabhat Jha as the Senior Editor. The following individual involved in review of your submission has agreed to reveal their identity: Rachel Daniels (Reviewer #2).

The reviewers have discussed the reviews with one another and the Reviewing Editor has drafted this decision to help you prepare a revised submission.

Summary:

The authors adapt a previously published mathematical model of *Plasmodium falciparum* transmission to consider the impact of rapid diagnostic tests (RDTs) on the prevalence of *pfhrp2* gene deleted mutants. Using data from the Democratic Republic of Congo they estimate the prevalence of *pfhrp2*-deleted mutants prior to RDT introduction and identify the effects of malaria prevalence and frequency of people seeking treatment on the rate of spread on *pfhrp2*-deleted mutants. Using parasite prevalence and treatment coverage estimates the model is used to identify areas of Sub-Saharan Africa of concern that should be prioritized for surveillance.

Essential revisions:

Broadly, the reviewers agreed this paper addressed an important subject, used an appropriate modelling framework and was well-presented. However the methods were described with insufficient detail to enable a full understanding of what was done. It was also felt that the authors needed to do more to demonstrate that their conclusion about geographical areas of high concern are robust to alternative assumptions. We expand on these concerns below.

1) The Materials and methods section does not precisely define the actual model used. Instead an analogous deterministic model is described. This is problematic as there is not a single corresponding stochastic model. The authors need make it much clearer what they actually did by specifying precisely the stochastic model they actually used (there are no space limitations so this should be possible). Some aspects of the model also don't appear to be directly analogous to the defined d.e. model. For example, the equations in subsection “*P. falciparum* transmission model” do not allow for superinfection, but the text states that the actual model does, but there is no information on how this is done, what multiplicity of infection is allowed for etc. Another example: in the Materials and methods section it is stated that "each u term represents the time during which immunity cannot be boosted further after a previous boost ", which is straightforward to define for a stochastic model (assuming the authors mean a constant u – do they?), but doesn't have an obvious correspondence with the given PDEs. It is also not made clear in the Materials and methods section what the relationship is between this model and the models of Griffin et al., 2014,2015,2016. This should be clarified.

Additionally, assumptions made for the admin 1 predictions are not clear. For geographic predictions was epsilon 1 at all times when no RDTs are used? What about slow-uptake of coverage of RDT use (or was it assumed that once RDTs are introduced there is immediate 100% coverage)? These assumptions have implications on the selection and on the conclusions about areas of concern for surveillance.

2) It would be a great help to non-modellers and improve readability to outline the key assumptions of the model (without equations) in the Introduction.

3) In the supplement, it would add transparency if some level of validation was provided that the model produces output that matches the prev/incidence by country/admin 1 as previously used by this group

4) It appears that only one clinical case can occur per infection. Does this mean only fully curative treatment is assumed (no sub-curative due to patient non-adherence, low dosing, lack of system compliance/resistance/fake drugs)? This should be clarified and the implications of this considered in the Discussion.

5) In the Results section and Discussion – "using the baseline frequency estimate of 6% prior to RDT" How do areas of concern change if this assumption was relaxed? The caveats around assuming a starting prevalence of 6% everywhere have been addressed in the Discussion, but in a somewhat simplified manner "Fourthly, extrapolating the starting frequency of *pfhrp2*-deletion strains from the DRC across the rest of SSA is a clear oversimplification; however, in the absence of similar datasets, we feel it provides a reasonable first estimate". However, the work would be further strengthened if the authors undertook a simplified analysis to see if/how priority geographic areas for surveillance change if a lower or higher starting frequency of *pfhrp2* was assumed. Along the lines of the other investigations concerning assumptions around epsilon etc. This seems particularly important if the authors agree with their own statement "Thus our results should be interpreted not as predictions of the absolute levels of the gene deletion, but rather indicative of geographical areas in which surveillance should be focused."

6) Discussion section: Concerning assumptions of non-treated RDT negative cases and coverage of RDT usage -"However, the ratio of testing via microscopy versus RDT is likely to have decreased over this period, and hence our estimate of RDT use (which our model assumes is 100% from introduction) is likely too high". It is not clear that this, and adherence/lack of adherence to RDT treatment guidelines, would not have significant impact on the main conclusions. There is a strong argument for additional sensitivity analysis here.

7) Figure 4 and Discussion: the classification of areas into high, moderate, slight and marginal concern is based on the frequency of *pfhrp2*-deleted mutants. But how does this frequency correlate with prevalence and ability to detect? From the results, it appears that low prevalence areas with high treatment rates are likely to increase in *pfhrp2*-deleted mutant frequency first, but these areas may not correspond to areas that are easy to detect/sample from nor, more importantly, areas where clinical morbidity will increase as infections go untreated as they result in negative RDTs. Can the authors comment on this and address the impact on Pf prevalence in relative terms considering the dynamic feedback with treatment and transmission?

8) It appears that the selection pressure for deletion mutants arises solely as a result of differential treatment rates of patients who are clinically diseased as a result of malaria. However, in high prevalence areas, many of those treated for malaria may be parasite positive, have a clinical diagnosis of malaria, but have a non-malarial cause of fever. In some settings these non-malarial causes could account for the vast majority of patients treated for malaria (for example Crump JA, Morrissey AB, Nicholson WL, Massung RF, Stoddard RA, Galloway RL, et al. (2013) Etiology of Severe Non-malaria Febrile Illness in Northern Tanzania: A Prospective Cohort Study. PLoS Negl Trop Dis 7(9): e2324. doi:10.1371/journal.pntd.0002324). Intuitively these effects could substantially change the selective pressures acting on the deletion mutants. If this process has been ignored (which is the reviewers' understanding) at the very least there should be a good argument explaining why it can be neglected. Additional analysis allowing for the fact that many patients treated for malaria may have other causes of fever would be the preferred option.

9) A related question is what happens to the patients who do have clinical disease resulting from malaria, but who don't get treated because they are infected only with the *pfhrp2* deletion mutants (and don't have cross-reactivity from pfhrp3). It appears from the model description these patients will never get treated for the current infection. If this is correct, how clinically realistic is this? In practice wouldn't a patient who continues to show clinical signs of malaria get treated for malaria even in the absence of a positive test result?

[Editors' note: further revisions were requested prior to acceptance, as described below.]

Thank you for resubmitting your work entitled "Modelling the drivers of the spread of *Plasmodium falciparum hrp2* gene deletions in sub-Saharan Africa" for further consideration at *eLife*. Your revised article has been favorably evaluated by Prabhat Jha (Senior editor) and a Reviewing editor.

The manuscript has been improved but there are some remaining issues that need to be addressed before acceptance, as outlined below:

The authors have done a very good job in responding to the reviewers' comments, and the changes have greatly strengthened this work and have addressed the concerns raised.

The only outstanding issues for me are what I presume are typos in one or two equations (and in one case, a clarification is needed). These equations were not given in the original submission, but they are needed for a full description of the work.

Substantive concerns:

1) Something is wrong with the second equation in the Materials and methods section. First, this gives a biting rate that goes down with age, not up (contradicting the text which follows the third equation). Secondly it allows for negative biting rates (and will give them at default parameter values). Minus sign missing?

2) For the hazards for the first three transitions in subsection “Immunity and detection functions it wasn't clear to me why the period u_. (corresponding to the time when immunity cannot be further boosted after a previous boost) was accounted for by scaling the hazard rather than allowing it change with time. Wouldn't it be straightforward to specify a hazard of h_i if no previous boost in the previous u_B days for person i (or u_C or u_D) and 0 otherwise?

3) Two things that seem odd in the Kolmogorov equations in subsection “Stochastic Model Equations”: i) I can't make sense of the first term on the rhs (P_i(j,k,l,m,a,t). Is there a -mu missing here? ii) on the fifth line of these equations shouldn't the $(1-f_t) epsilon$ term be $(1-f_t epsilon)$ ?

[Editors' note: further revisions were requested prior to acceptance, as described below.]

Thank you for resubmitting your work entitled "Modelling the drivers of the spread of *Plasmodium falciparum hrp2* gene deletions in sub-Saharan Africa" for further consideration at *eLife*. Your revised article has been favorably evaluated by Prabhat Jha (Senior editor) and one reviewer, who is also a member of our Board of Reviewing Editors.

The manuscript has been improved and addressed the previous concerns but there are some remaining issues that need to be addressed before acceptance, as outlined below in the review.

Reviewer 1:

This revision has introduced a further change in the way the model is described. This is a good thing if it aids reproducibility and transparency. The one problem is that the new formulation is very difficult to make sense of, and the logic of the revised equation is not clear.

Some text should be added below these equations to explain in a less technical way what these terms on the r.h.s. represent. It may well be that this equation as it stands is correct. Or there could be one or more mistakes which need to be fixed.

Apart from this issue, I don't have any further concerns with the paper.

[Editors' note: further revisions were requested prior to acceptance, as described below.]

Thank you for resubmitting your work entitled "Modelling the drivers of the spread of *Plasmodium falciparum hrp2* gene deletions in sub-Saharan Africa" for further consideration at *eLife*. Your revised article has been evaluated by Prabhat Jha (Senior editor) and a Reviewing editor.

Unfortunately, the revisions have not sufficiently addressed the concerns raised in the previous review and the issues raised there still need to be addressed before acceptance. Given the lengthy back and forth on this review, we must insist that this will be the last opportunity we will allow for a successful response to the concerns expressed by the Reviewing editor.

In particular, a precise, intelligible description of the model is an essential requirement. While the new mathematical description of the model is precise, it is not fully intelligible. This may be because there are still errors in the description, or it may because sufficient motivation for equations has not been provided. To take just one example, on the sixth line of the equations in subsection “Stochastic Model Equations” there is a term that seems to correspond to bites from infectious mosquitos which do not result in infection (the bites occur at rate h_i, the EIR, and (1-b_i) is the probability of no infection given such a bite). Suppose a susceptible gets bitten, so j is S and t_k< t < t_k + u_b), so the function theta evaluates to 1. So in this case the operator b evaluates to the identify function, and plugging this back into the equation at the top of the page we learn that the rate P_i(S,…) is changing w.r.t time (keeping age constant) for susceptible hosts is (1-b_i) * h_i (..) * P_i(S,…).

Why should this be the case? Why are these non-infecting bites leading to increases in the susceptible population? This is not clear and is merely the first place where the equations in this subsection cannot be readily understood.

We propose two possible ways forward:

i) a detailed explanation of the equations is added, explaining how they represent the model going into a similar level of detail as the text above.

ii) the simulation model is described in pseudocode rather than in equations. We appreciate that R code is already provided on Github. However, R will not be universally readable (either by other researchers now or in the future who are not familiar with R [and the R language is itself evolving so R code that is valid now may not run in the future]). If the model is described in pseudocode that should allow the twin aims of precision and intelligibility.

Option ii) might be easier, and of course (since space is not an issue) both i) and ii) could be done together.

---

## [Author Response]

Essential revisions:Broadly, the reviewers agreed this paper addressed an important subject, used an appropriate modelling framework and was well-presented. However the methods were described with insufficient detail to enable a full understanding of what was done. It was also felt that the authors needed to do more to demonstrate that their conclusion about geographical areas of high concern are robust to alternative assumptions. We expand on these concerns below.1) The Materials and methods section does not precisely define the actual model used. Instead an analogous deterministic model is described. This is problematic as there is not a single corresponding stochastic model. The authors need make it much clearer what they actually did by specifying precisely the stochastic model they actually used (there are no space limitations so this should be possible).

We thank the reviewers for raising the need for further clarity pertaining to the precise numerical implementation of the stochastic model. We have re-drafted the Materials and methods to fully specify the stochastic model in a way that we hope will still be accessible to those who do not follow the model equations. To do this, we have removed the deterministic descriptions and provided a list of transitions for the infection states in the human model and their associated hazards. In the section describing the acquisition and loss of naturally-acquired immunity, we have similarly provided a table with the transitions and associated hazards. We also explicitly include the standard equation to translate a continuous-time hazard rate into a daily transition probability since we implement the stochastic model on a daily time step. For mathematical clarity, we formalise this by presenting the Kolmogorov forward equation for the human infection states.

Some aspects of the model also don't appear to be directly analogous to the defined d.e. model. For example, the equations in subsection “P. falciparum transmission model” do not allow for superinfection, but the text states that the actual model does, but there is no information on how this is done, what multiplicity of infection is allowed for etc.

We thank the reviewers for highlighting the need for extra clarity and information here. In our revisions the superinfection process is clearly included in the equations and the tables of transitions.

In response to comments regarding how the multiplicity of infection is allowed for, we are deeply grateful for the reviewers bringing this to our attention as an additional paragraph detailing this was erroneously omitted in the earlier submission. This paragraph has been added in to the end of the PfHRP2 dynamics section of the transmission model, and reads as follows:

“Individuals that clear infection lose all strains, and infected individuals clear a random strain at rate *nr_C_*, where n is the total number of strains and r_C_ is the rate at which one strain is cleared in a monoinfected individual, i.e. *1/(d_A_ + d_U_)*. This introduces a carrying capacity on the number of strains an individual can be infected with, which scales with the transmission intensity. The multiplicity of infection and strain profile of an infected individual have no effect on the disease outcome except when the use of RDTs is introduced.”

We have not added the strains into the forward equations as this would make them very difficult to read but feel that this text description is now sufficiently clear to be reproducible.

Another example: in the Materials and methods section it is stated that "each u term represents the time during which immunity cannot be boosted further after a previous boost ", which is straightforward to define for a stochastic model (assuming the authors mean a constant u – do they?), but doesn't have an obvious correspondence with the given PDEs.

This was originally in the deterministic equivalent equations but has now been included in the forward equations. To make this clearer in the text this sentence now reads as follows: “Each u term represents the time during which immunity cannot be boosted further after a previous boost and each d term represents the duration of immunity. In the stochastic implementation of the model used here, each immunity term increases by 1 for an individual whenever that individual receives an infectious bite or is infected respectively, if the previous boost to I_B_, I_CA_ and I_D_ occurred more than u_B_, u_C_, and u_D_ days earlier respectively.”

It is also not made clear in the Materials and methods section what the relationship is between this model and the models of Griffin et al., 2014,2015,2016. This should be clarified.

The basic disease model is the same as in *Griffin et al.,* 2016, which is an amalgamation of previous iterations of this model in 2014 and 2015. The model is extended here to include the dynamics of *pfhrp2*-deleted parasites, through tracking the infection history of individuals and introducing assumptions about the impact of these parasites upon HRP2-based RDT guided treatment decisions. To make this clearer we have added the following to the beginning of the opening paragraph of the Materials and methods: “An individual-level stochastic model was developed to simulate the transmission dynamics of *Plasmodium falciparum*. The model is based upon previous modelling efforts, and is described in full here before describing the extensions made with regards to PfHRP2 dynamics, and defining the parameters used and their sources.”

Additionally, assumptions made for the admin 1 predictions are not clear. For geographic predictions was epsilon 1 at all times when no RDTs are used? What about slow-uptake of coverage of RDT use (or was it assumed that once RDTs are introduced there is immediate 100% coverage)? These assumptions have implications on the selection and on the conclusions about areas of concern for surveillance.

We thank the reviewers for these comments and have added the following to the Materials and methods to increase the clarity detailing the simulation assumptions for the Africa admin 1 predictions: “These simulations considered populations of 100,000 individuals that were simulated for 20 years from 2010 to 2030, with the introduction of RDTs assumed across all regions in 2010.” We fully agree with the implications related to the latter half of this comment, and have extended our sensitivity analysis accordingly and increased our discussion surrounding this within the Discussion section. Full details of this extension are made in response to comment 4 within the essential revisions where this point is raised again.

2) It would be a great help to non-modellers and improve readability to outline the key assumptions of the model (without equations) in the Introduction.

We fully agree with these comments, and have extended the Introduction to highlight the assumptions detailing RDT nonadherence, microscopy use, fitness costs and the impact of non-malarial fevers. This was added as follows:

“In addition, we conduct sensitivity analyses to characterise the influence of assumptions within our model concerning adherence to RDT-guided treatment decisions, the use of microscopy-based diagnostic testing, fitness costs associated with the mutant parasite and the impact of non-malarial fevers upon the selective advantage of *pfhrp2* gene deletions.”

3) In the supplement, it would add transparency if some level of validation was provided that the model produces output that matches the prev/incidence by country/admin 1 as previously used by this group

The model is set up to exactly match the prevalence at a given time-point in any location through scaling the mosquito density M_V_. For transparency we have now included an additional figure supplement (Figure 4—figure supplement 1), which shows the aligned model output compared to the estimates of prevalence within SSA produced by the Malaria Atlas Project. We have made reference to this extra figure in the final paragraph of the Results, the opening two sentences of which now read: “Finally, using the baseline frequency estimate of 6% prior to RDT introduction, we explored 1000 different prevalence and treatment seeking rates spanning the range of estimates of the PfPR and treatment levels across sub-Saharan Africa (SSA) in 2010 (see Figure 4—figure supplement 2). The model output was aligned with these estimates by first administrative units (see Figure 4—figure supplement 1), which enabled us to project the potential spread of the mutant strain and its impact on RDT-guided treatment (see Video 1).”

The relationship between prevalence and clinical incidence is identical to our previous outputs since the model is an exact replica of this in the absence of RDT mutations.

4) It appears that only one clinical case can occur per infection. Does this mean only fully curative treatment is assumed (no sub-curative due to patient non-adherence, low dosing, lack of system compliance/resistance/fake drugs)? This should be clarified and the implications of this considered in the discussion.

We agree that sub-curative treatment is an important consideration but this is not captured in our current model. We have made clear our assumption about all treatment being fully-curative within the substantial rewrite of the second paragraph of the Materials and methods section. Sub-curative treatment would likely slow the rate of selection, along with a number of other factors detailed in this section, which we have introduced into the 3^rd^ paragraph of the Discussion section.

5) In the Results section and Discussion – "using the baseline frequency estimate of 6% prior to RDT" How do areas of concern change if this assumption was relaxed? The caveats around assuming a starting prevalence of 6% everywhere have been addressed in the Discussion, but in a somewhat simplified manner "Fourthly, extrapolating the starting frequency of pfhrp2-deletion strains from the DRC across the rest of SSA is a clear oversimplification; however, in the absence of similar datasets, we feel it provides a reasonable first estimate". However, the work would be further strengthened if the authors undertook a simplified analysis to see if/how priority geographic areas for surveillance change if a lower or higher starting frequency of pfhrp2 was assumed. Along the lines of the other investigations concerning assumptions around epsilon etc. This seems particularly important if the authors agree with their own statement "Thus our results should be interpreted not as predictions of the absolute levels of the gene deletion, but rather indicative of geographical areas in which surveillance should be focused."

We thank the reviewers for this useful comment. To address this we considered 3 additional different starting frequencies of *pfhrp2-*deletion (3%, 4.5% and 7.5%) with an assumed probability of a clinical case seeking treatment, who is only infected with *pfhrp2*-deleted mutants, producing a positive RDT result (ϵ)equal to 0.25. The final frequency of *pfhrp2-*deletion across SSA was subsequently recorded and admin level 1 regions ranked accordingly, as shown in the new Figure 4—figure supplement 4. This extra sensitivity analysis is discussed within the penultimate paragraph of the Discussion as follows:

“Fourthly, extrapolating the starting frequency of *pfhrp2*-deletion strains from the DRC across the rest of SSA is a clear oversimplification; however, in the absence of similar datasets, we feel it provides a reasonable first estimate. […]Despite the expected changes in the final frequency of *pfhrp2*-deletion in these settings, the overall pattern of areas with the highest selection-driven increase in *pfhrp2*-deletion remains the same.”

6) Discussion section: Concerning assumptions of non-treated RDT negative cases and coverage of RDT usage -"However, the ratio of testing via microscopy versus RDT is likely to have decreased over this period, and hence our estimate of RDT use (which our model assumes is 100% from introduction) is likely too high". It is not clear that this, and adherence/lack of adherence to RDT treatment guidelines, would not have significant impact on the main conclusions. There is a strong argument for additional sensitivity analysis here.

We agree fully with this comment, and have conducted a series of additional sensitivity analyses in response to this comment and comment 8 within the essential revisions in relation to the impact of non-malarial fevers. With regards to non-adherence and the use of microscopy-based diagnosis we conducted a similar suite of simulations to those in Figure 1, in which we incorporated a probability of diagnosis via microscopy occurring and a probability of non-adherence to RDT results occurring. The methodology for this is expanded within the new second paragraph within the Materials and methods, which also explores the introduction of non-malarial fevers. The results of this are introduced in the new third paragraph of the Results section.

These factors, as expected, caused a decrease in the selective pressure (Figure 1—figure supplement 3). This decrease, along with the decrease due to potential fitness costs sit in opposition to an increase observed due to non-malarial fever (discussed further in response to comment 8). As such we would want to stress our acknowledgment of these opposing factors, as discussed at greater length in the third paragraph of the Discussion, and highlight again a need for further data to allow our predictions of geographical areas of HRP2 Concern to be refined. However, with the data currently available we believe our model gives a good overall representation of how these factors might average out, as represented by the model’s ability to predict the relationship within DRC.

7) Figure 4 and Discussion: the classification of areas into high, moderate, slight and marginal concern is based on the frequency of pfhrp2-deleted mutants. But how does this frequency correlate with prevalence and ability to detect? From the results, it appears that low prevalence areas with high treatment rates are likely to increase in pfhrp2-deleted mutant frequency first, but these areas may not correspond to areas that are easy to detect/sample from nor, more importantly, areas where clinical morbidity will increase as infections go untreated as they result in negative RDTs. Can the authors comment on this and address the impact on Pf prevalence in relative terms considering the dynamic feedback with treatment and transmission?

We thank the reviewers for these comments, as the interaction between false-negative test results, treatment and transmission intensity has important clinical implications resulting from changes in the frequency of *pfhrp2*-deleted mutants. Firstly, to demonstrate this we looked at how the malaria prevalence in Figure 1 was changing as a result of increases in *pfhrp2-*deletion due to the introduction of RDT-guided treatment decisions (Figure 1—figure supplement 1). These simulations predict that as a result of increasing *pfhrp2-*deletion, which will be greatest at the lowest transmission intensities, the prevalence of malaria increases due to an increased number of false-negative test results. We have added these findings to the end of the second paragraph of the Results section as follows: “The prevalence of malaria within Figure 1 was also observed to increase after RDT introduction (Figure 1—figure supplement 1), with the greatest increase in lower transmission settings resulting from untreated infections due to false-negative RDT results”. We have also incorporated these implications within the fourth paragraph of the Discussion, reiterating the need for careful interpretation of these results as follows: “Similarly, these results should not be interpreted as predictions of the precise negative impact on malaria prevalence as a result of increase gene deletions (Figure 1—figure supplement 1), but illustrative of the potential impact of false-negative test results upon prevalence and the importance of alternative diagnostic methods (Figure 1—figure supplement 3).”

We also agree that low transmission areas where this effect is greatest may not be the easiest to sample from, for example due to the absence of adequate sampling infrastructure as a result of the areas historical low transmission. If these areas, however, have a high risk of *pfhrp2-*deletion due to a high frequency of treatment it seems likely that the infrastructure in these settings would be sufficient. We also agree that areas with low prevalence and high treatment coverage will have fewer cases so the impact on clinical morbidity will likely be lower due to stronger health care systems and the ability to notice these false-negative systems. For example, a recent report from Eritrea (Berhane A, Russom M, Bahta I, Hagos F, Ghirmai M, Uqubay S. 2017. Rapid diagnostic tests failing to detect *Plasmodium falciparum* infections in Eritrea: an investigation of reported false negative RDT results. Malar J 16: 105. doi:10.1186/s12936-017-1752-9), shows that although the effect on morbidity may not be severe, due to their extensive use of RDTs from 2007, within settings with a high treatment seeking rate, a substantial prevalence of *pfhrp2-*deletion has now been observed. Conversely, settings with the highest clinical cases will also have the highest multiplicities of infection, so are less likely to experience negative RDT results as clinical cases are unlikely to be monoinfected.

8) It appears that the selection pressure for deletion mutants arises solely as a result of differential treatment rates of patients who are clinically diseased as a result of malaria. However, in high prevalence areas, many of those treated for malaria may be parasite positive, have a clinical diagnosis of malaria, but have a non-malarial cause of fever. In some settings these non-malarial causes could account for the vast majority of patients treated for malaria (for example Crump JA, Morrissey AB, Nicholson WL, Massung RF, Stoddard RA, Galloway RL, et al. (2013) Etiology of Severe Non-malaria Febrile Illness in Northern Tanzania: A Prospective Cohort Study. PLoS Negl Trop Dis 7(7): e2324. doi:10.1371/journal.pntd.0002324). Intuitively these effects could substantially change the selective pressures acting on the deletion mutants. If this process has been ignored (which is the reviewers' understanding) at the very least there should be a good argument explaining why it can be neglected. Additional analysis allowing for the fact that many patients treated for malaria may have other causes of fever would be the preferred option.

We believe this a very important question, and are indebted to the reviewers for raising it. As mentioned in response to comment 6, we conducted additional sensitivity analyses in response to this comment. In brief, we attempted to estimate an annual rate of non-malarial fever a different age ranges to capture the increased variability in fevers at lower ages. This was conducted by using all available Demographic Health Surveys that recorded data regarding previous fevers, and had sufficiently low rates of treatment sought from drug peddlers or hawkers. This was then averaged across discrete age brackets, with more age brackets at the lower ages and combined with the estimated proportion of fevers that are non-malarial fevers from across Africa in 2007 (Gething PW, Kirui VC, Alegana VA, Okiro EA, Noor AM, Snow RW. 2010. Estimating the number of paediatric fevers associated with malaria infection presenting to Africa’s public health sector in 2007. PLoS Med 7. doi:10.1371/journal.pmed.1000301). We elected to use this estimate as we hoped it would be more representative of the range of non-malarial fevers across Africa.

This inclusion of non-malarial fevers created a substantial increase in the selective pressure (Figure 1—figure supplement 4), and if we had incorporated the proportion of fevers that are non-malarial from the cited study from Tanzania, this increase would have been even greater. This increase was notably greater than individual decreases to the selective pressure from potential microscopy use or non-adherence to RDT results, the results of which are introduced in the new third paragraph of the Results section.

As mentioned earlier, we want to reiterate our acknowledgment of these opposing factors, and highlight that the presented model managed to capture the trend shown in DRC, and yielded a selection pressure that was very similar to that observed when considering the intermediate levels of the model assumptions together within our sensitivity analysis (Figure 1—figure supplement 5). We are aware, however, that in other regional settings these factors might lead to potentially a slower or faster rate of selection-driven increase in *pfhrp2*-deletion. An example of this can be seen in the recent report from Eritrea (Berhane A, Russom M, Bahta I, Hagos F, Ghirmai M, Uqubay S. 2017. Rapid diagnostic tests failing to detect *Plasmodium falciparum* infections in Eritrea: an investigation of reported false negative RDT results. Malar J 16: 105. doi:10.1186/s12936-017-1752-9), where RDTs were deployed in 2007 within settings with a high treatment seeking rate and a substantial prevalence of *pfhrp2-*deletion has now been observed. We also have mentioned, however, that our model over predicted the likely prevalence of *pfhrp2*-deletion in Senegal, which again highlights that our results should be interpreted not as predictions of the absolute levels of the gene deletion, but rather indicative of geographical areas in which surveillance should be focused. We have included a summary of these arguments within bot the third and fourth paragraphs of the Discussion, included the recent Eritrean study as well as a recent study from Rwanda within Table 1, and would like to thank the reviewers again for raising this important point.

9) A related question is what happens to the patients who do have clinical disease resulting from malaria, but who don't get treated because they are infected only with the pfhrp2 deletion mutants (and don't have cross-reactivity from pfhrp3). It appears from the model description these patients will never get treated for the current infection. If this is correct, how clinically realistic is this? In practice wouldn't a patient who continues to show clinical signs of malaria get treated for malaria even in the absence of a positive test result?

We thank the reviewers for raising this point. In the model, a clinically diseased individual who does not receive treatment will move to an asymptomatic infection on average after 5 days, before potentially recovering via a sub-patent infection, or being infected again at which point they may be treated. We agree that it is plausible that patients who show continued clinical symptoms of malaria in the absence of a positive test result may be treated, either through an alternative diagnostic method being used, or due to nonadherence to the test result. These potential routes to treatment are considered in response to earlier comments made by the reviewers. The extensions to the third paragraph of the Discussion raises this point, explaining how these routes may capture the clinical realism of patients being treated without a positive RDT result due to their continued presentation of clinical symptoms of malaria.

[Editors' note: further revisions were requested prior to acceptance, as described below.]

The only outstanding issues for me are what I presume are typos in one or two equations (and in one case, a clarification is needed). These equations were not given in the original submission, but they are needed for a full description of the work.Substantive concerns:1) Something is wrong with the second equation in the Materials and methods section. First, this gives a biting rate that goes down with age, not up (contradicting the text which follows the third equation). Secondly it allows for negative biting rates (and will give them at default parameter values). Minus sign missing?

Thank you for highlighting this omission. The minus sign was missing and has since been added.

2) For the hazards for the first three transitions in subsection “Immunity and detection functions it wasn't clear to me why the period u_. (corresponding to the time when immunity cannot be further boosted after a previous boost) was accounted for by scaling the hazard rather than allowing it change with time. Wouldn't it be straightforward to specify a hazard of h_i if no previous boost in the previous u_B days for person i (or u_C or u_D) and 0 otherwise?

These hazard rates have now been removed, as the extended Kolmogorov equations now better describe the acquisition of immunity using a series of commutative integral operators with indicator functions to represent the delay in further boosting.

3) Two things that seem odd in the Kolmogorov equations in subsection “Stochastic Model Equations”: i) I can't make sense of the first term on the rhs (P_i(j,k,l,m,a,t). Is there a -mu missing here? ii) on the fifth line of these equations shouldn't the $(1-f_t) epsilon$ term be $(1-f_t epsilon)$ ?

Thank you for highlighting these. The first term on the rhs has been removed in the extended equations, and the epsilon term has been moved inside the bracket.

[Editors' note: further revisions were requested prior to acceptance, as described below.]Reviewer 1:This revision has introduced a further change in the way the model is described. This is a good thing if it aids reproducibility and transparency. The one problem is that the new formulation is very difficult to make sense of, and the logic of the revised equation is not clear.Some text should be added below these equations to explain in a less technical way what these terms on the r.h.s. represent. It may well be that this equation as it stands is correct. Or there could be one or more mistakes which need to be fixed.Apart from this issue, I don't have any further concerns with the paper.

We agree the extensions made to the stochastic equations could make the reproducibility of the model more difficult. However, we have included these equations in response to earlier requests made by the reviewers to include a single model that specified precisely the stochastic model. The extensions made in these revised equations now allow the period of time in which immunity is not boosted to be described formulaically, which addresses the reviewer’s earlier essential comments concerning the issue of describing an analogous deterministic model.

We do, however, believe the lines of the model mentioned by the reviewer are hard to interpret. Additional text has been added after these equations that now reads: “These functions allow the fixed periods of time in which immunities are not boosted after a previous boost to be included within the stochastic equations, while also allowing superinfection events to be incorporated.” We also want to mention again that the simulation model has been made publicly available within the Github repository listed within the Materials and methods to help reproducibility.

[Editors' note: further revisions were requested prior to acceptance, as described below.]

In particular, a precise, intelligible description of the model is an essential requirement. While the new mathematical description of the model is precise, it is not fully intelligible. This may be because there are still errors in the description, or it may because sufficient motivation for equations has not been provided. To take just one example, on the sixth line of the equations in subsection “Stochastic Model Equations” there is a term that seems to correspond to bites from infectious mosquitos which do not result in infection (the bites occur at rate h_i, the EIR, and (1-b_i) is the probability of no infection given such a bite). Suppose a susceptible gets bitten, so j is S and t_k< t < t_k + u_b), so the function theta evaluates to 1. So in this case the operator b evaluates to the identify function, and plugging this back into the equation at the top of the page we learn that the rate P_i(S,…) is changing w.r.t time (keeping age constant) for susceptible hosts is (1-b_i) * h_i (..) * P_i(S,…).Why should this be the case? Why are these non-infecting bites leading to increases in the susceptible population? This is not clear and is merely the first place where the equations in this subsection cannot be readily understood.We propose two possible ways forward:i) a detailed explanation of the equations is added, explaining how they represent the model going into a similar level of detail as the text above.ii) the simulation model is described in pseudocode rather than in equations. We appreciate that R code is already provided on Github. However, R will not be universally readable (either by other researchers now or in the future who are not familiar with R [and the R language is itself evolving so R code that is valid now may not run in the future]). If the model is described in pseudocode that should allow the twin aims of precision and intelligibility.Option ii) might be easier, and of course (since space is not an issue) both i) and ii) could be done together.

In response to these valid concerns we have decided to carry out both suggested ways forward. In response to the request for model pseudocode we have written out the full model using a mathematical style syntax with detailed annotations that describing and explaining the rationale behind each line of code, including both a pseudocode syntax and colour key to aid understanding of the syntax. This has been included as a supplementary file (Supplementary file 1), and has been referenced to within the main text within the “Model code availability and parameter values” paragraph of the Materials and methods. We have added this as a supplementary file due to the considerable length of the pseudocode model, and to hopefully preserve the monospaced font and colour syntax, which aid the intelligibility of the pseudocode.

In addition we have added four new paragraphs following the stochastic model equations that provides a detailed explanation of the motivation for each line of equations. This new text reads as follows:

“The stochastic model equations detailed above can be explained as follows. The first line is the total time derivative of Pi (j,k,tk,l,tl,m,tm,cm,a,t). The next four lines describe the flows into states S, A, U and P due to progression through infection states. […] The last line shows the removal of individuals from the population through death, balancing the inflow from the previous line.”